# *GATA6* mutations in hiPSCs inform mechanisms for maldevelopment of the heart, pancreas, and diaphragm

Arun Sharma[1,2,3†], Lauren K Wasson[1,4†], Jon AL Willcox[1], Sarah U Morton[1,5], Joshua M Gorham[1], Daniel M DeLaughter[1], Meraj Neyazi[1,6], Manuel Schmid[1,7], Radhika Agarwal[1], Min Young Jang[1], Christopher N Toepfer[1,8,9], Tarsha Ward[1], Yuri Kim[1], Alexandre C Pereira[1,10], Steven R DePalma[1], Angela Tai[1], Seongwon Kim[1], David Conner[1], Daniel Bernstein[11], Bruce D Gelb[12], Wendy K Chung[13], Elizabeth Goldmuntz[14], George Porter[15], Martin Tristani-Firouzi[16], Deepak Srivastava[17], Jonathan G Seidman[1], Christine E Seidman[1,4,18]*, Pediatric Cardiac Genomics Consortium

[1]Department of Genetics, Harvard Medical School, Boston, United States; [2]Smidt Heart Institute, Cedars-Sinai Medical Center, Los Angeles, United States; [3]Board of Governors Regenerative Medicine Institute, Cedars-Sinai Medical Center, Los Angeles, United States; [4]Howard Hughes Medical Institute, Harvard Medical School, Boston, United States; [5]Division of Newborn Medicine, Boston Children's Hospital, Boston, United States; [6]Hannover Medical School, Hannover, Germany; [7]Deutsches Herzzentrum München, Technische Universität München, Munich, Germany; [8]Division of Cardiovascular Medicine, Radcliffe Department of Medicine, University of Oxford, Oxford, United Kingdom; [9]Wellcome Centre for Human Genetics, University of Oxford, Oxford, United Kingdom; [10]Laboratory of Genetics and Molecular Cardiology, Heart Institute, Medical School of University of Sao Paulo, Sao Paulo, Brazil; [11]Department of Pediatrics, Stanford University School of Medicine, Stanford, United States; [12]Department of Genetics and Genomic Sciences, Icahn School of Medicine at Mount Sinai, New York, United States; [13]Department of Medicine, Columbia University Medical Center, New York, United States; [14]Department of Pediatrics, The Perelman School of Medicine, University of Pennsylvania, Philadelphia, United States; [15]Department of Pediatrics, University of Rochester Medical Center, Rochester, United States; [16]Division of Pediatric Cardiology, University of Utah School of Medicine, Salt Lake City, United States; [17]Gladstone Institutes, San Francisco, United States; [18]Cardiovascular Division, Department of Medicine, Brigham and Women's Hospital, Boston, United States

*For correspondence:
cseidman@genetics.med.harvard.edu

†These authors contributed equally to this work

Competing interests: The authors declare that no competing interests exist.

**Abstract** Damaging *GATA6* variants cause cardiac outflow tract defects, sometimes with pancreatic and diaphragmic malformations. To define molecular mechanisms for these diverse developmental defects, we studied transcriptional and epigenetic responses to *GATA6* loss of function (LoF) and missense variants during cardiomyocyte differentiation of isogenic human induced pluripotent stem cells. We show that GATA6 is a pioneer factor in cardiac development, regulating *SMYD1* that activates *HAND2,* and *KDR* that with *HAND2* orchestrates outflow tract formation. LoF variants perturbed cardiac genes and also endoderm lineage genes that direct *PDX1* expression and pancreatic development. Remarkably, an exon 4 *GATA6* missense variant, highly associated with extra-cardiac malformations, caused ectopic pioneer activities, profoundly diminishing *GATA4, FOXA1/2,* and *PDX1* expression and increasing normal retinoic acid signaling

that promotes diaphragm development. These aberrant epigenetic and transcriptional signatures illuminate the molecular mechanisms for cardiovascular malformations, pancreas and diaphragm dysgenesis that arise in patients with distinct *GATA6* variants.

## Introduction

Congenital heart disease (CHD), the leading birth defect worldwide that occurs in approximately 1% of newborns (*van der Linde et al., 2011*), comprises a range of structural malformations arising during embryonic development (*Zaidi and Brueckner, 2017*). Contemporary treatments for CHD enables survival into adulthood, but many patients have ongoing medical problems related to extracardiac malformations and/or neurodevelopmental disabilities (*Brickner et al., 2000*; *Marino et al., 2012*). Understanding the developmental mechanisms that cause co-occurrence of these congenital anomalies is expected to provide better patient care, improve predictive risk assessments, and potentially uncover therapeutic targets.

Heart development is an intricately regulated process driven by genetic, epigenetic, and biomechanical events that form a complex, multi-chambered, muscular organ containing cardiomyocytes and other cell lineages (*Litviňuková et al., 2020*). Early in cardiogenesis, the first heart field creates the left ventricle and portions of the atria, while the second heart field gives rise to the right ventricle and outflow tract, with contributions from neural crest cells (*Buckingham et al., 2005*). Multiple transcription factors orchestrate particular components of these developmental processes (*Srivastava and Olson, 2000*), as evidenced by regional defects in CHD patients with damaging mutations in particular genes (*Fahed et al., 2013*; *Jin et al., 2017*; *Sifrim et al., 2016*). For example, dominant human loss of function (LoF) variants in *TBX5*, *NKX2-5*, and *GATA4* that are highly expressed in the first and second heart fields cause atrial and ventricular septal defects (*Basson et al., 1997*; *Benson et al., 1999*; *Garg et al., 2003*). *GATA6*, while expressed in the first heart field (*Morrisey et al., 1996*), plays particularly critical roles in the developing second heart field (*Molkentin, 2000*) and in recruitment of cardiac neural crest lineages (*Lepore et al., 2006*) that together shape the cardiac outflow tract. Consistent with these developmental functions, CHD patients with *GATA6* mutations have a striking preponderance of outflow tract malformations (*Gharibeh et al., 2018*; *Kodo et al., 2009*; *Maitra et al., 2010*). *GATA6* also is critical for endodermal development (*Fisher et al., 2017*), and some CHD patients with damaging *GATA6* variants also have pancreatic agenesis, congenital diaphragmatic hernia (*Yu et al., 2014*) or other abdominal malformations (*Chao et al., 2015*; *De Franco et al., 2013*; *Shi et al., 2017*).

Developmental transcription factors work in concert to exert network-level effects on organogenesis (*Luna-Zurita et al., 2016*): physical interactions of NKX2-5, TBX5, and GATA4 proteins modulate the expression of other cardiac genes (*Garg et al., 2003*; *Hiroi et al., 2001*; *Bruneau et al., 2001*; *Maitra et al., 2009*) while GATA6 and GATA4 proteins together promote pancreatic development (*Chao et al., 2015*; *De Franco et al., 2013*; *Shi et al., 2017*). Notably, members of the GATA family of transcription factors serve as endodermal pioneer factors that participate with FOXA, also a pioneer factor, to engage and open chromatin and recruit additional transcriptional activators (*Fisher et al., 2017*; *Zaret and Carroll, 2011*). Thus, we expect that understanding the molecular networks in which *GATA6* participates will help to elucidate the mechanisms by which human mutations cause defects in morphogenesis of the heart cardiac other organs.

A traditional approach for studying developmental mechanisms relies on gene disruption in model organisms. While often informative, prior studies of mice with one inactivated *Gata6* allele have either subtle dysmorphic aortic valves (*Gharibeh et al., 2018*) or no CHD (*Lepore et al., 2006*) and notably lack pancreatic agenesis or congenital diaphragmatic hernia that occur in human CHD patients with damaging *GATA6* variants (*Chao et al., 2015*; *De Franco et al., 2013*; *Shi et al., 2017*). Mice with biallelic *Gata6* inactivation have deficits in early visceral endoderm formation resulting in nonviable embryos (*Morrisey et al., 1996*; *Zhao et al., 2005*), which may explain the absence of homozygous *GATA6* null alleles in humans, but provide limited insights into organ morphogenesis. The introduction of human CHD mutations using CRISPR/Cas9 (*De Franco et al., 2013*) into isogenic human induced pluripotent stem cell-derived cardiomyocytes (hiPSC-CMs) provides an alternative model system suited for analyzing developmental consequences, as differentiation of these cells activate transcriptional networks that regulate early in vivo human cardiogenesis

(*DeLaughter et al., 2016*; *Li et al., 2016*). Moreover, isogenic hiPSC-CMs containing distinct variants can illuminate variant-specific transcriptional patterns.

We employed this approach to study human de novo GATA6 variants, identified by whole-exome sequencing (WES) of CHD patients. We demonstrate the graded transcriptional effects of *GATA6* heterozygous and homozygous LoF variants during hiPSC-CM differentiation. We also examine the transcriptional requirements for a specific arginine residue encoded by *GATA6* exon four that is recurrently mutated in unrelated CHD patients. Combining these data with Assay for transposase-accessible chromatin using sequencing (ATAC-seq) (*Buenrostro et al., 2015*; *Corces et al., 2017*) and GATA6 ChIP-seq analyses, we demonstrate that GATA6 is a pioneer factor for cardiac development. We define direct and indirect transcriptional responses to *GATA6* variants. Integrating these datasets with clinical phenotypes observed in CHD patients with pathogenic *GATA6* variants, we demonstrate how disrupted molecular programs cause aberrant development of the cardiac outflow tract, pancreas, and diaphragm.

## Results

### Identification of CHD patients with *GATA6* LoF and missense variants

Among >4000 CHD patients enrolled and studied by WES through National Heart, Lung, and Blood Institute's Pediatric Cardiac Genomics Consortium (PCGC) (*Homsy et al., 2015*; *Jin et al., 2017*), we identified nine heterozygous de novo variants: four LoF (LoF) and five damaging missense variants in *GATA6* (*Figure 1A*). The congenital anomalies in these patients were consistent with previously recognized roles for *GATA6* in developing the cardiac outflow tract, arterial-ventricular valves, and posterior brachial arches that form aorta and pulmonary vessels (*Gharibeh et al., 2018*; *Laforest and Nemer, 2011*; *Losa et al., 2017*). Among PCGC CHD patients and 61 previously reported CHD patients with pathogenic *GATA6* variants (*Figure 1B*) there were a preponderance of outflow tract malformations, including persistent truncus arteriosus, double-outlet right ventricle, tetralogy of Fallot, as well as aortic and pulmonary valve and septation defects (*Kelly, 2012*). Some but not all of these patients also had extra-cardiac phenotypes including pancreatic agenesis, congenital diaphragmatic hernia, and neurodevelopmental deficits.

We considered whether the distribution of these 70 damaging *GATA6* variants (61 published, 9 PCGC) across the 595 encoded GATA6 amino acids correlated with clinical phenotypes (*Figure 1*; *Supplementary file 1A, B*). All variants (43 missense, 27 LoF, including eight recurrent variants) caused CHD. Extra-cardiac phenotypes occurred in 29/70 (41%) patients, and more often with LoF (18/27) than missense (11/43) variants (p<0.001). Neurocognitive dysfunction occurred in 13 patients (18.5%). Pancreatic agenesis/hypoplasia or congenital diaphragmatic hernia occurred in 20/70 patients (28.5%) and more frequently with *GATA6* LoF (14/27) than missense (9/43) variants (p=0.001).

Among 43 *GATA6* missense variants, 11 variants altered residues in the DNA-binding zinc finger (ZF) domain encoded by exon 4 (amino acids 435–477), significantly more than expected by chance (p=0.0004). Nine of these 11 patients with exon four missense variants had pancreatic agenesis or congenital diaphragmatic hernia, but none of 32 patients with missense variants located elsewhere (p=1.1e-6). Within exon 4, recurrent missense mutations altered the basic arginine residue 456. Computational modeling of this domain (PyMOL software) positioned residue 456 alongside a polar residue (asparagine 466) in close proximity to DNA (*Bates et al., 2008*; *Figure 1C*). Substitution of a non-polar glycine at residue 456 (R456G) is predicted to disrupt these interactions, and potentially alter GATA6 binding to DNA.

### Generation of *GATA6* LoF and *GATA6*$^{R456G/R456G}$ hiPSCs using CRISPR/Cas9

We created *GATA6* LoF hiPSCs using two independent guide RNAs (gRNAs) targeting exon 2 (*Figure 2—figure supplement 1A*) that were transfected with Cas9 endonuclease into an early passage healthy hiPSC line PGP1 (*Lee et al., 2009*). Targeted hiPSCs were subcloned and *GATA6* variants were confirmed by next-generation and Sanger sequencing of PCR-amplified fragments (METHODS). Four independent mutant hiPSC lines were obtained: two carry a heterozygous 1 bp insertion (*GATA6*$^{+/-}$, chr18:19,752,124–19,752,124, A:TA) and two have a homozygous 1 bp deletion

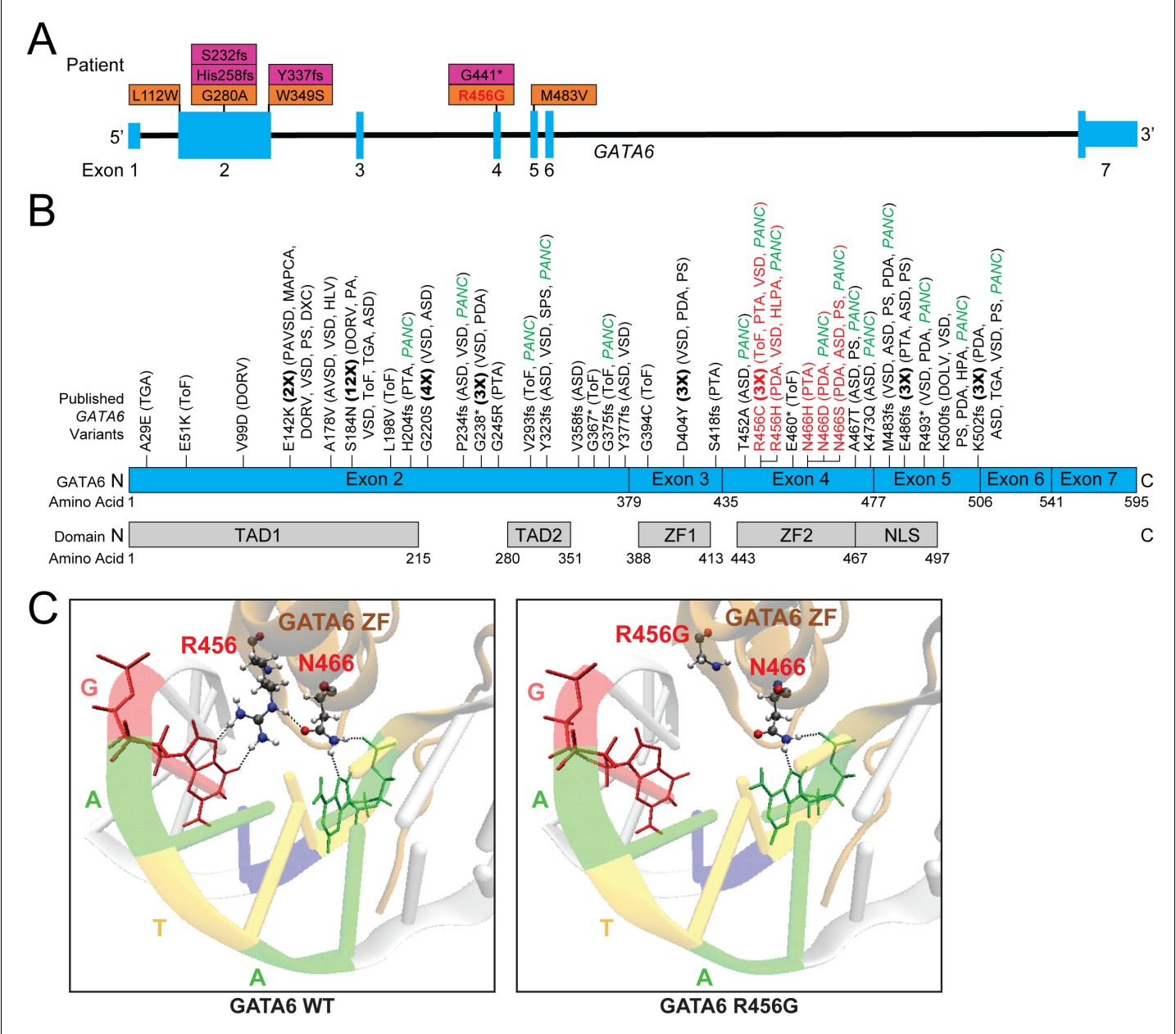

**Figure 1.** Genetic information and clinical phenotypes of individuals with *GATA6* variants. **A**) Schematic of *GATA6* gene and locations of PCGC *GATA6* variants (LoF-purple, missense-orange). (**B**) Previously described (see *Supplementary file 1*) *GATA6* variants (R456 and N466 variants highlighted in red), and GATA6 protein domains. TAD: Topologically associating domain, ZF: zinc finger, NLS: nuclear localization signal. TGA: Transposition of the Great Arteries, ToF: Tetralogy of Fallot, DORV: Double-Outlet Right Ventricle, DOLV: Double-Outlet Left Ventricle, DXC: Dextrocardia, VSD: Ventricular Septal Defect, HLV: Hypoplastic Left Ventricle, PA: Pulmonary Atresia, ASD: Atrial Septal Defect, PTA: Persistent Truncus Arteriosus, SPS: Supravalvular Pulmonary Stenosis, HLPA: Hypoplastic Left Pulmonary Artery, PS: Pulmonary Stenosis, HPA: Hypoplastic Pulmonary Artery, PANC: Pancreatic Agenesis (**C**) Model of GATA6 DNA-binding domain bound to major groove of DNA indicating the location of amino acid residue 456. Left panel: GATA6 residues R456 and N466 normally interact with each other via hydrogen bonding (dashed lines) and with target G base and second A base in the GATA motif via hydrogen bonding, respectively (dashed lines). Right panel: Replacing the arginine (R) residue at position 456 with a glycine (G) residue alters normal molecular interactions by disrupting the hydrogen bonds.

(*GATA6⁻/⁻*, chr18:19,752,123–19,752,124, TA:T). Using similar strategies, we transfected a gRNA targeting exon four with a single-stranded DNA oligonucleotide to serve as a template for homology-directed repair and generated two hiPSC lines with homozygous missense variant R456G (*GATA6^{R456G/R456G}*; *Figure 2—figure supplement 1B*). No lines with a heterozygous R456G variant

were obtained. In parallel we produced *GATA6⁺/⁻*, *GATA6⁻/⁻*, and *GATA6^{R456G/R456G}* variants in a PGP1 hiPSC line that carried expressed green fluorescent protein (GFP) fused to endogenous cardiac troponin T alleles (TNNT2-GFP; *Figure 2—figure supplement 1C*).

## Differentiation of *GATA6* mutant hiPSCs into hiPSC-CMs

Wildtype (WT), *GATA6⁺/⁻*, *GATA6⁻/⁻*, and *GATA6^{R456G/R456G}* hiPSCs were processed for differentiation into cardiomyocytes (hiPSC-CMs) by modulation of the Wnt signaling pathway and subsequent metabolic selection via glucose deprivation (*Sharma et al., 2018b*). This protocol yields hiPSC-CMs that express first and second heart field genes (*Zhang et al., 2019*). As previous single-cell RNA-sequencing (RNA-Seq) of cardiomyocytes isolated from developing mouse hearts (*DeLaughter et al., 2016*) identified peak *Gata6* expression in cardiac progenitors and early cardiomyocytes, we studied hiPSC-CMs at differentiation days 4 and 8, which approximate these in vivo developmental stages. GATA6 protein expression and nuclear localization were reduced in *GATA6⁺/⁻* compared to isogenic WT lines and absent in *GATA6⁻/⁻* lines (*Figure 2—figure supplement 1D,E*). *GATA6⁺/⁻* hiPSCs, unlike WT hiPSCs, exhibited mono-allelic *GATA6* expression, suggesting nonsense-mediated decay of RNAs transcribed from targeted alleles.

We assessed sarcomere production as a measure of cardiomyocyte differentiation in the TNNT2-GFP lines. *GATA6⁺/⁻* hiPSC-CMs had weaker fluorescent signal than WT cells, while *GATA6⁻/⁻* lines expressed no fluorescence at baseline or during the differentiation protocol (*Figure 2A,B*). Consistent with these data, at differentiation day eight when contracting sarcomeres were present in WT cells, fewer independent differentiation rounds of *GATA6⁺/⁻* hiPSC-CMs contained beating

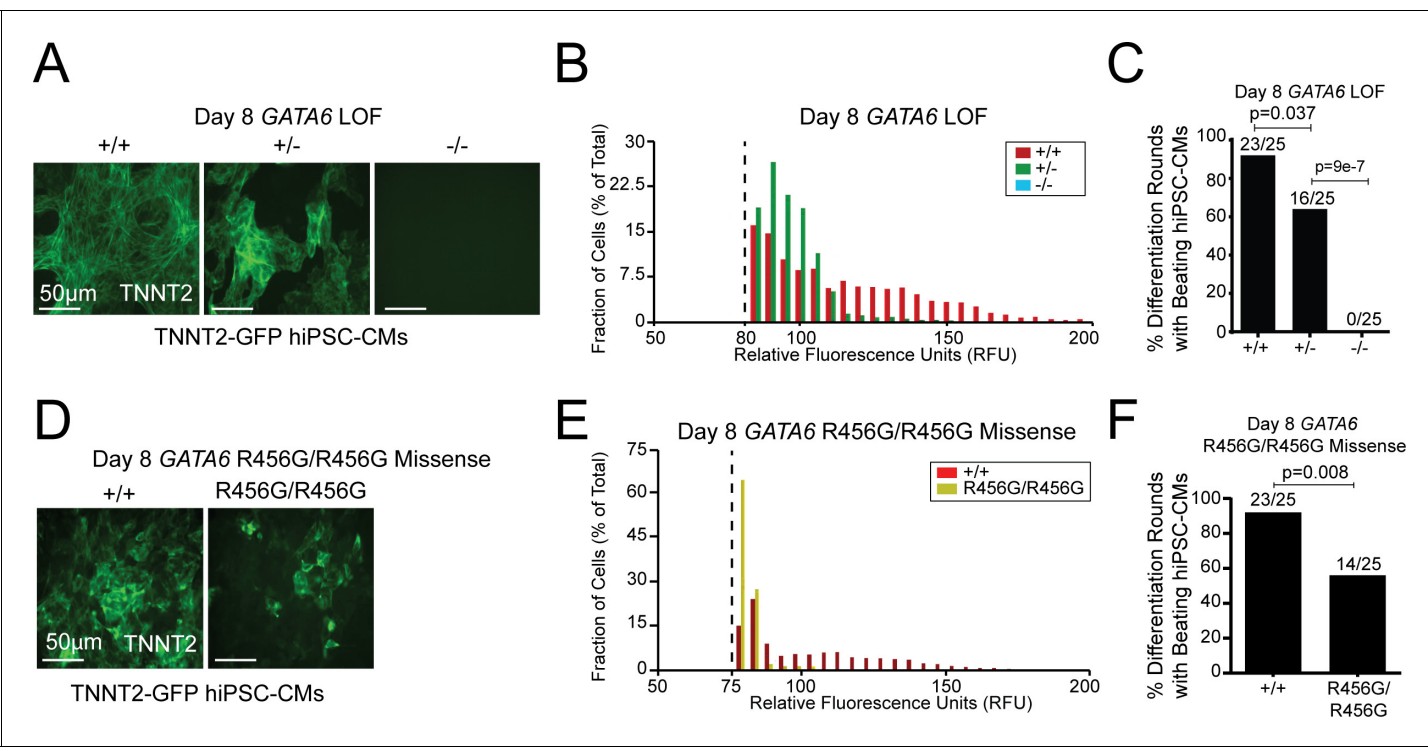

**Figure 2.** *GATA6* mutant hiPSCs exhibit hiPSC-CM differentiation defects. (A) *GATA6* variants in a TNNT2-GFP reporter line showed reduced (*GATA6⁺/⁻*) or absent (*GATA6⁻/⁻*) GFP-tagged sarcomeres in comparison to WT cells. (B) Distribution of dissociated GFP-TNNT2-*GATA6* mutant cells assessed using the Countess system with a GFP filter cube (METHODS). (C) Number of *GATA6* LoF mutant differentiation cultures (n = 25 per genotype) with beating hiPSC-CMs. (D) The *GATA6^{R456G/R456G}* variant has reduced expression of GFP-tagged sarcomeres. (E) Fluorescence distribution of differentiated GFP-TNNT2 *GATA6^{R456G/R456G}* cells assessed using the Countess system with a GFP filter cube (see Materials and methods). (F) Number of GFP-TNNT2 *GATA6^{R456G/R456G}* differentiation cultures (n = 25) with beating day eight hiPSC-CMs. All lines were studied at differentiation day 8. Significance was assessed using Student's t-test.

The online version of this article includes the following figure supplement(s) for figure 2:

**Figure supplement 1.** Sequence and phenotype characterization of *GATA6* mutant hiPSCs.

sarcomeres and no *GATA6*[-/-] cells showed sarcomeres or spontaneous beating (*Figure 2C*). Analogous studies of *GATA6*[R456G/R456G] hiPSC-CMs (*Figure 2D–F*) showed a fluorescence signal comparable to that of *GATA6*[+/-] hiPSC-CMs and spontaneous beating cells in approximately ~60% of differentiation rounds.

## Transcriptional analysis of *GATA6* LoF hiPSC-CMs

We assessed transcriptional responses to altered *GATA6* levels throughout hiPSC-CM differentiation by bulk RNA-Seq analyses of cultures at days 0, 4, 8, 12, and 30 (*Figure 3*, *Figure 2—figure supplement 1F*, *Figure 3—figure supplement 1*, *Supplementary file 2*), and by single-cell RNA-Seq on days 4 and 8 (*Figure 4*) that precedes metabolic selection and enriches cultures for cardiomyocytes. All cells were differentiated, processed, and harvested in parallel. RNA-Seq data was aligned and processed to limit potential batch effects, and clustered using methods implemented in DESEQ2 (bulk) or Seurat (single cell) (METHODS). We observed consistency between bulk and single-cell expression data. In addition, principal component analyses (PCA) (*Figure 3A*, *Supplementary file 3*) of independent, genotype-identical lines demonstrated close clustering of RNA expression, indicating that *GATA6* genotype and differentiation stage largely accounted for differences in gene expression.

At day 4 of differentiation, RNA-Seq of WT cells showed expression of both pluripotent stem cell markers (e.g., *POU5F1* encoding OCT4) and transcriptional modulators associated with early cardiomyocyte differentiation (*Figures 3B,C* and *4A,B*). These primordial cardiomyocytes expressed *SMYD1* (SET and MYND domain-containing protein-1), a nuclear histone methyl-transferase involved in remodeling chromatin and sarcomere assembly (*Li et al., 2011*), Tbox transcription factors *TBX5* and *TBX20*, two of the earliest markers of cardiac development (*Bruneau et al., 2001*; *Takeuchi et al., 2005*), and *MEF2C*, an essential transcription factor for sarcomere assembly and function (*Lin et al., 1997*). From day eight onward, the expression of stem cell marker genes was extinguished while expression of cardiomyocyte differentiation transcripts increased (*Figures 3C* and *4C,D*).

Our protocol for cardiomyocyte differentiation also yielded subpopulations of cells expressing endodermal genes. Day 4 WT cells expressed hepatocyte nuclear factor family members (*HNF1*, *HNF4A*, *HNF4B*) that are depleted in *Gata6*-null mice (*Morrisey et al., 1998*), *FOXA1* and *FOXA2*, transcription factors that activate expression of the pancreas/duodenum homeobox-1 gene *PDX1*, which is essential for pancreas development and β islet cell differentiation (*Gao et al., 2008*; *Gerrish et al., 2001*; *Lee et al., 2019*; *Zhou et al., 2008*) and other endodermal markers such as *SOX17* (*Wang et al., 2011*; *Figure 4A,B* and *Supplementary file 2*). *PDX1* was expressed through differentiation day 12 in WT lines, but not after metabolic enrichment for cardiomyocytes.

*GATA6*[+/-] cells at day 4 of differentiation had lower transcript levels of primordial cardiomyocyte genes compared to WT cells (*Figure 3B,C*) and gene ontology (GO) analyses inferred that pathways involved in cardiac muscle contraction, development, and chamber organization were reduced. Notably, the expression of *SMYD1*, which regulates the expression of *HAND2*, a critical transcriptional regulator of the second heart field (*Gottlieb et al., 2002*; *Laurent et al., 2017*) that gives rise to the outflow tract, was strikingly diminished (*Figure 3B–D*, *Supplementary file 2*). Similarly, expression of multiple *HAND2* network genes was also lower than in WT cells (*Figure 3C,D*), including *GATA* gene family members, *ISL1*, *MEF2C*, and prototypic cardiac transcription factors *MEIS-1*, *NKX2-5*, *TBX5*, and *TBX20*.

Prompted by the transcriptional signatures in *GATA6*[+/-] cells, we compared the heart malformations in 54 PCGC patients with damaging variants in 11 *HAND2*-network and second heart field genes (*Supplementary file 1C*) with patients who have *GATA6* mutations. Forty of these 54 patients also had outflow tract malformations, including all (n = 5) CHD patients with damaging de novo variants in *TBX20*.

*GATA6*[+/-] cells also identified dysregulation of other gene programs involved in forming the outflow tract. The transient peak expression in WT cells of *HOXA1*, *HOXB1*, and *MSX1*, which regulate the induction and expression of critical molecules involved in specifying neural crest cell development (*Makki and Capecchi, 2011*; *Simões-Costa and Bronner, 2015*; *Tvrdik and Capecchi, 2006*), remained low throughout differentiation of *GATA6*[+/-] cells. Notably, *HOXB1* also participates in specifying endoderm destined for pancreatic and other abdominal cell lineages (*Huang et al., 2002*). Expression was also diminished of key molecules that couple outflow tract myocardium with

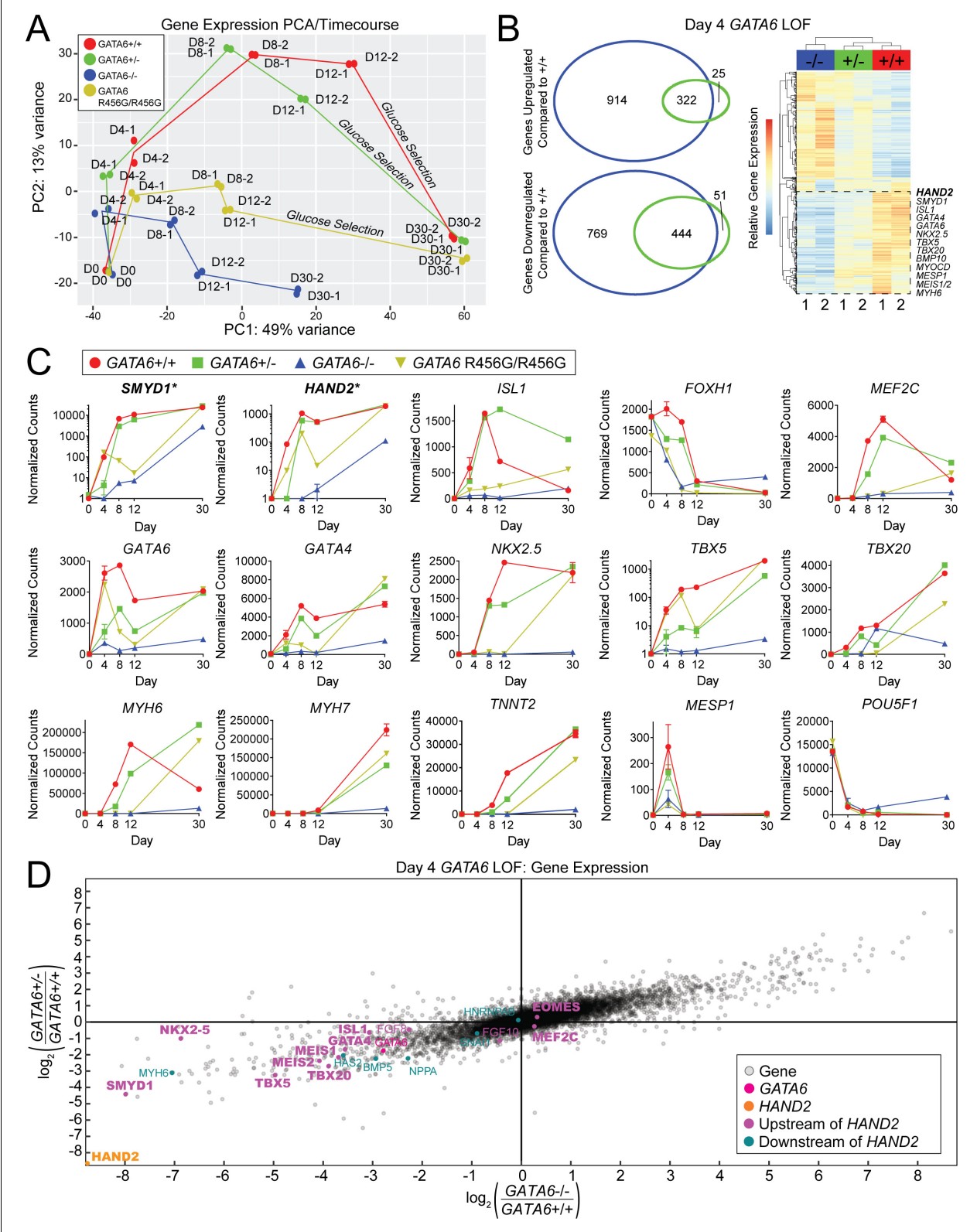

**Figure 3.** *GATA6* mutant cells exhibit downregulation of second heart field-related genes during hiPSC-CM differentiation. **A)** Gene expression principal component analysis (PCA) of day 0–30 WT (*GATA6*$^{+/+}$), *GATA6*$^{+/-}$, *GATA6*$^{-/-}$, and *GATA6*$^{R456G/R456G}$ hiPSC-CMs. RNA-Seq samples were harvested in duplicate for all time points. (**B**) Venn Diagrams (left) and heat map (right) of day 4 *GATA6*$^{+/-}$ and *GATA6*$^{-/-}$ cells. In the heatmap, red indicates upregulated genes whereas blue represents downregulated genes. Samples are in duplicate. Selected second heart field genes are shown.

*Figure 3 continued on next page*

*Figure 3 continued*

(C) Expression data in normalized counts for second heart field-related genes (top row), cardiac developmental transcription factors (middle row), sarcomere, and other selected genes (bottom row) during differentiation of *GATA6* mutant hiPSC-CMs. Data represented as mean ± SD. Note that *SMYD1* and *HAND2* graphs are plotted with logarithmic scale. (D) Gene expression scatterplot illustrating downregulation of expression of *HAND2* upstream and downstream gene network in day 4 *GATA6*$^{+/-}$ and *GATA6*$^{-/-}$ cells. X-axis, log2 fold-change of gene expression in *GATA6*$^{-/-}$ cells relative to WT. Y-axis, log2-fold-change of gene expression in *GATA6*$^{+/-}$ cells relative to WT. Canonical cardiac development and the second heart field genes are bolded.

The online version of this article includes the following figure supplement(s) for figure 3:

**Figure supplement 1.** *GATA6* mutant cells exhibit an upregulation in epithelial-to-mesenchymal transition, neurodevelopmental, and neural crest-related genes.

developing vascular beds, including *KDR* (encoding vascular endothelial growth factor (VEGF) receptor-2) and the VEGF co-receptor, *NRPI* (*Supplementary file 2*). Human mutations in *HOXA1* (*Tischfield et al., 2005*) and *KDR* (*Reuter et al., 2019*) alleles cause cardiac outflow tract malformations. Our data infers a critical role for *GATA6* in orchestrating *SMYD1-HAND2, VEGF*, and neural crest network interactions during outflow tract morphogenesis.

Day 8 *GATA6*$^{+/-}$ cells expressed many cardiomyocyte transcripts at levels found in WT lines, but single-cell RNA-Seq analyses demonstrated some persistent differences including reduced expression of *MYH7*, a defining transcript of mature cardiomyocytes (*Figure 3C*, *Figure 4C,D* and *Figure 2—figure supplement 1F*). Genes encoding other sarcomere proteins were reduced through day 12 in *GATA6*$^{+/-}$ cells and abnormalities remained after metabolic enrichment for cardiomyocytes (*Supplementary file 2*). Compared to WT, day 30 *GATA6*$^{+/-}$ cardiomyocytes had 10-fold higher expression levels of *ISL1* and 4-fold higher levels of fetal myosin transcripts (*MYH6*), suggesting developmental immaturity of *GATA6* mutant cardiomyocytes.

Endodermal lineage genes were variably misexpressed in differentiating *GATA6*$^{+/-}$ cells. In comparison to WT cells, transcript levels were lower for *GATA4, HNF1*, and *HNF4A* but normal for *FOXA1* and *FOXA2* (*Supplementary file 2*). The day 12 peak expression of *PDX1* in WT cells was absent in *GATA6*$^{+/-}$ cells.

*GATA6*$^{+/-}$ cells also misexpressed transcription factors genes involved in diaphragm development. In day 8 cells transcripts encoding *NR2F2* (encoding a retinoic acid (RA) responsive transcription factor) were 2 to 3-fold higher while expression of *ZFPM2* (encoding a family member of the Friend of GATA (FOG) transcription factors) was half of WT levels. As damaging variants in these genes cause congenital diaphragmatic hernia (*Kardon et al., 2017*), these data imply that *GATA6* haploinsufficiency causes diaphragmatic dysgenesis by disrupting *NR2F2* and *ZFPM2* gene programs (*Supplementary file 2*).

*GATA6*$^{-/-}$ hiPSC had profound deficits in cardiomyocyte differentiation (*Figure 3*, *Figure 2—figure supplement 1F*, *Supplementary file 2*). Transcripts associated with early mesoderm specification including *MESP1* and transcription factors associated with primordial cardiomyocytes were more depressed than in *GATA6*$^{+/-}$ cells. These findings support prior observations that shRNA knockdown of *GATA6* reduces hiPSC-CM differentiation capacity (*Yoon et al., 2018*). Given the absence of emerging cardiomyocytes, metabolic selection to further enrich for this lineage was not performed.

The expression of endodermal genes in *GATA6*$^{-/-}$ hiPSCs was also abnormal. Throughout differentiation *GATA4* transcripts were 10-fold reduced, *FOXA1* and *FOXA2* were transiently expressed only at day 4, and *HNF4A* and *PDX1* expression were extinguished (*Supplementary file 2*). Dysregulated expression of *NR2F2* (increased) and *ZFPM2* (decreased) was also observed.

Single-cell RNA-Seq (*Figure 4C,D*) identified two *GATA6*$^{-/-}$ populations. One population maintained high expression of the stem cell (SC) marker *POU5F1* throughout differentiation day 8, while the other (FB) had increased expression of fibroblast markers (*COL3A1, IGFBP7*), epithelial to mesenchymal transition markers (*SNAIL1/2, MMP2, VIM*) (*Figure 3—figure supplement 1A*), and neural differentiation genes (*SHH, ZEB1/2, NCAM2*) (*Figure 3—figure supplement 1A,B*). Together these data inferred that *GATA6*$^{-/-}$ cells, lacking the normal signals involved in specifying cardiomyocyte and endoderm lineages, adopted alternative differentiation programs.

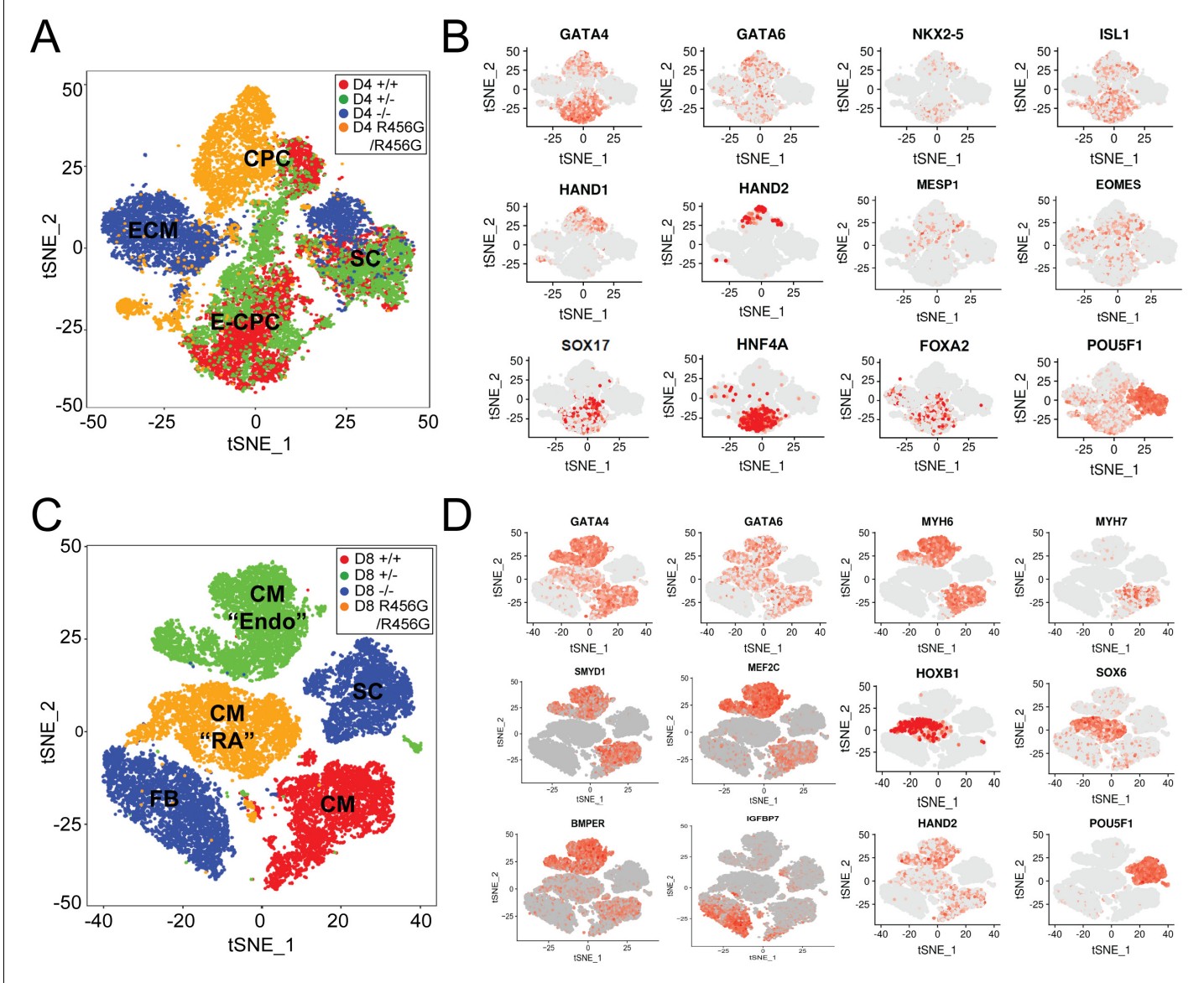

**Figure 4.** Single-cell transcriptional analysis of *GATA6* mutant hiPSC-CMs during differentiation. (A) tSNE of single-cell RNA-Seq of day 4 *GATA6* mutant hiPSC-CMs identified four clusters. Labels reflect marker gene expression: SC, Stem cell; CPC, cardiac progenitor cells; E-CPC, cardiac progenitors enriched with endoderm markers; ECM, endodermal-like cells enriched for extracellular matrix proteins. (B) Examples of marker gene expression in clustered hiPSC-derived cells. CPC cells express mesodermal factors (*MESP1, EOMES*) as well as cardiac transcription factors (*GATA6, GATA4, ISL1, NKX2.5, HAND1,* and *HAND2*). SCs expressed OCT4 (*POU5F1*). E-CPCs expressed *GATA6, SOX17, HNF4A,* and *FOXA2*. (C) tSNE clustering of single-cell RNA-Seq of day 8 *GATA6* mutant hiPSC-CMs identified five clusters. Labels reflect marker gene expression: CM, cardiomyocytes; CM ('RA'), cardiomyocytes with increased RA-signaling pathway genes; SC, Stem Cell; CM ('Endo'), cardiomyocytes with enrichment in endothelial genes; FB; fibroblast-like cells (D) Examples of marker gene expression in clustered cells. SC expressed OCT4 (*POU5F1*). CMs expressed sarcomere protein genes (*MYH6, MYH7*), *SMYD1*, a CM-specific histone methyl-transferase, and *HAND2,* a second heart field transcription factor. CMs (RA) also expressed retinoic acid pathway genes (*SOX6, HOXB1*). CMs (Endo) have upregulated endothelial cell gene expression (*MEF2C, BMPER*), while FB cells expressed ECM markers (*IGFBP7*).

The online version of this article includes the following figure supplement(s) for figure 4:

**Figure supplement 1.** retinoic acid inhibitor (RA inh) treatment of *GATA6^{R456G/R456G}* hiPSCs partially rescues cardiac progenitor gene expression.

## Transcriptional analysis of *GATA6*<sup>R456G/R456G</sup> hiPSC-CMs

Parallel analyses of isogenic *GATA6*<sup>R456G/R456G</sup> lines showed shared and distinct transcriptional profiles from *GATA6*<sup>+/-</sup> or *GATA6*<sup>-/-</sup> cells. At days 4–12, primordial cardiomyocyte transcripts (*GATA4*, *HAND2*, and *SMYD1*) in *GATA6*<sup>R456G/R456G</sup> lines were expressed at levels midway between *GATA6*<sup>+/-</sup> and *GATA6*<sup>-/-</sup> cells, but markedly below levels in WT cells (*Figures 3C* and *4*, and *Supplementary file 2*).

Distinctive transcription profiles in differentiating *GATA6*<sup>R456G/R456G</sup> cells (days 4–12) suggested aberrant RA signaling (*Figure 4C,D*, *Supplementary file 2*). *ALDH1A2* transcripts (encoding the enzyme that converts RA from retinaldehyde) were 50-fold higher than WT cells and the expression of *RARA*, *RARB* (RA receptors A and B), and *STRA6* (receptor for retinol uptake) were increased 2 to 3-fold. Transcripts encoding targets of RA signaling were also increased. *HOXB1,* which contains two RA-responsive elements, including one 6.5 kb 3' of coding sequences that is critical for foregut expression (*Huang et al., 2002*), was 100-fold increased at differentiation day 4 – a marked contrast to the depressed levels observed in *GATA6*<sup>+/-</sup> cells.

RA signaling modulates neural crest cell development (*Martínez-Morales et al., 2011*) and at day 4 *GATA6*<sup>R456G/R456G</sup> cells had increased *PAX3* and *PAX7* expression (100-fold and 3-fold, respectively; *Supplementary file 2*, *Figure 3—figure supplement 1B*). *HOXA1* (6-fold increased) and *HOXB* contribute to activating expression of *ZIC1* (8-fold increased) (*Jaurena et al., 2015*; *Makki and Capecchi, 2011*; *Tvrdik and Capecchi, 2006*) that with *PAX3/7* induce and specify a neural crest gene program. *PAX3* also promotes neural crest migration into the cardiac outflow tract development (*Conway et al., 1997*) and like *STRA6*, (*Coles and Ackerman, 2013*) and contributes to diaphragm development (*Stuelsatz et al., 2012*) by specifying paraxial mesoderm (*Magli et al., 2019*). Aberrant RA signaling may contributed to dysregulation of other developmental regulators of diaphragm formation, including *NR2F2* (~8-fold increased) and *ZFPM2* (5- to 12-fold decreased).

We further assessed activation of RA signaling by comparing RNA-Seq analyses of day 4 differentiating WT and *GATA6*<sup>R456G/R456G</sup> cells that were cultured with or without an inhibitor of the ALDH1A2 enzyme (METHODS). Treated WT cells (*Figure 4—figure supplement 1A*) had significantly altered expression (p<0.05, 1.5-fold), decreasing the expression of transcripts assigned to Gene Ontology terms for heart and muscle cell differentiation and increasing endodermal fate specification and Wnt signaling transcripts. Treated *GATA6*<sup>R456G/R456G</sup> cells (*Figure 4—figure supplement 1B*) had increased expression of transcripts assigned to Gene Ontology terms for myoblast differentiation, circulatory system development and anatomical structures. Notably, the expression of 25% of genes in treated *GATA6*<sup>R456G/R456G</sup> cells (*Figure 4—figure supplement 1C–F*) had levels found in untreated WT cells (*HOXB1* and *FOXH1*) or normalized levels, approaching those in untreated WT cells (*HOXA1*, *HAND2*, *RARB*, *TBX20*).

*GATA6*<sup>R456G/R456G</sup> cells had aberrant expression of other endoderm genes, including markedly lower transcripts of *HNF* gene family members and striking loss (≥100-fold below WT) of *FOXA1* and *FOXA2* expression (*Supplementary file 2*). Biallelic deletion of *FOXA2* in human stem cells prevents pancreatic specification (*Lee et al., 2019*). *GATA6*<sup>R456G/R456G</sup> cells also had increased expression of *SOX6*, which represses *PDX1*-dependent transcriptional activation of the insulin gene, *INS* (*Iguchi et al., 2005*). The cumulative effect of these deficits could account for the extinguished expression of *PDX1* and contribute to the high association of pancreatic agenesis with heterozygous *GATA6*<sup>R456G</sup> mutations (*Figure 1*).

Despite multiple transcriptional abnormalities, metabolic enrichment yielded *GATA6*<sup>R456G/R456G</sup> cardiomyocytes (day 30) with prototypic gene expression. However, like *GATA6*<sup>+/-</sup> hiPSC-CMs, *GATA6*<sup>R456G/R456G</sup> hiPSC-CMs exhibited a higher *MYH6:MYH7* transcript ratio than WT, indicating immaturity.

## GATA6 is a pioneer factor for cardiac development

We studied potential mechanisms by which *GATA6* regulated gene transcription by performing Assays for Transposase-Accessible Chromatin sequencing (ATAC-seq), GATA6 chromatin immunoprecipitation sequencing (ChIP-seq) and Hypergeometric Optimization of Motif EnRichment (HOMER) analyses at day 4 (*Figure 5*, *Figure 6*) and day 8 (*Supplementary file 4*). ChIP-seq of *GATA6*<sup>-/-</sup> cells yielded very few peaks above background (data not shown), confirming antibody specificity.

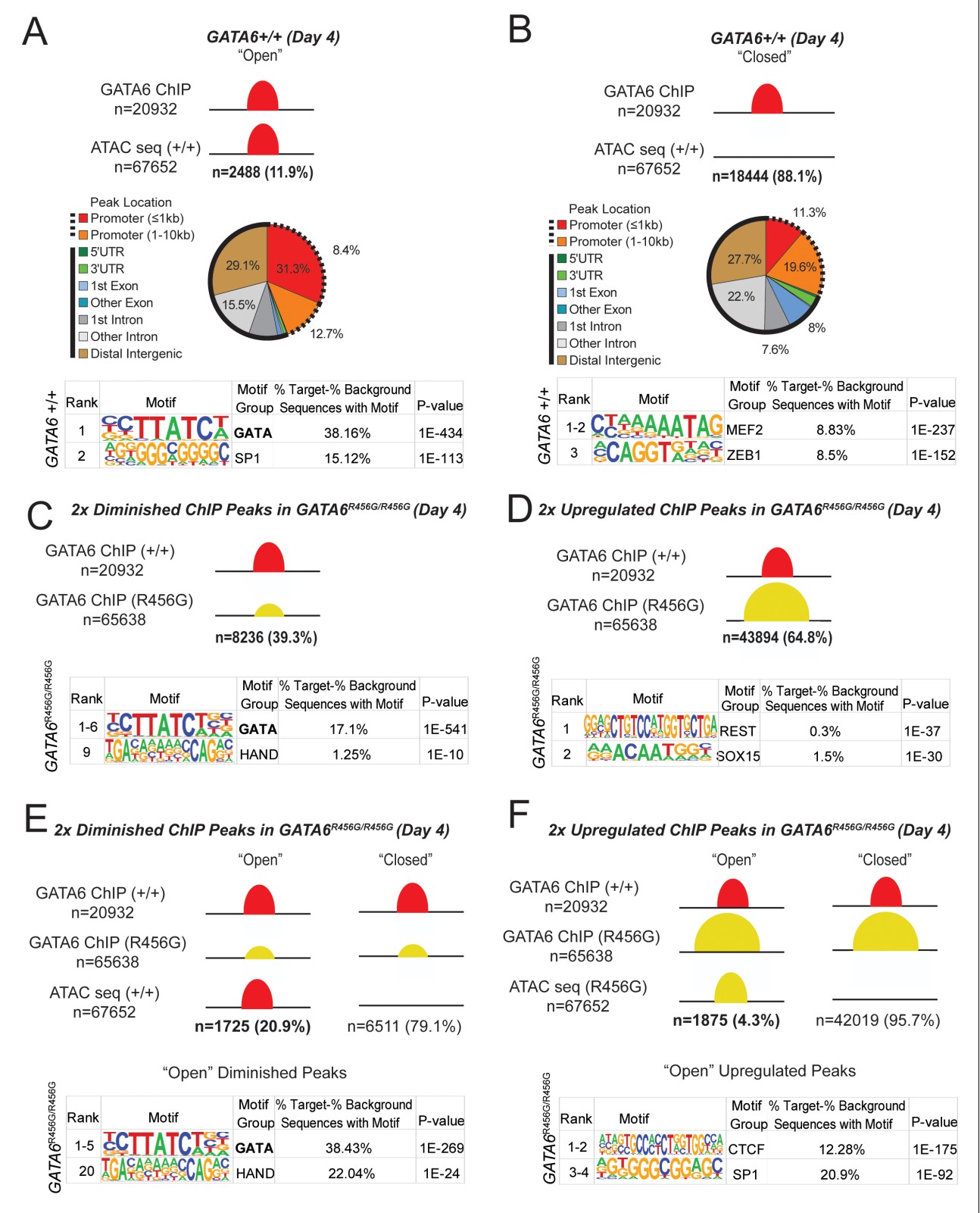

**Figure 5.** GATA6 is a pioneer factor for cardiac development. ATAC-seq and GATA6 ChIP-seq were performed in WT and mutant day 4 hiPSCs and overlapped to assess GATA6 direct binding to open vs. closed chromatin. (A) Approximately 12% of GATA6 ChIP-seq peaks overlapped with an ATAC-seq peak and were characterized as 'Open'. The genomic location of these ChIP-seq peaks in open chromatin were characterized with respect to gene bodies, and DNA-binding motif enrichment was performed using HOMER analysis (METHODS). When GATA6 binds to open regions of chromatin,

*Figure 5 continued on next page*

*Figure 5 continued*

peaks are enriched for the GATA motif. (B) The remaining 88% of GATA6 ChIP-seq peaks were characterized as 'Closed'. The genomic location of these ChIP-seq peaks were characterized with respect to gene bodies, and DNA-binding motif enrichment was performed using HOMER analysis. When GATA6 binds to closed regions of chromatin, peaks are enriched for the MEF2 motif. (C) Of the 20932 WT GATA6 ChIP-seq peaks, 39.3% were reduced in *GATA6^{R456G/R456G}* cells (adjusted p<1e-4, two fold). These peaks were enriched for the GATA and HAND2 binding motifs by HOMER analysis. (D) Of the 67652 *GATA6^{R456G/R456G}* ChIP-seq peaks, 64.8% were upregulated in *GATA6^{R456G/R456G}* cells (adjusted p<1e-4, two fold). These peaks were enriched for the REST and SOX binding motifs by HOMER analysis. (E) Peaks diminished in *GATA6^{R456G/R456G}* cells were overlapped with WT ATAC-seq data to establish chromatin accessibility. Almost 21% of peaks diminished in *GATA6^{R456G/R456G}* cells were in open chromatin regions; these peaks were enriched for the GATA and HAND motifs. (F) Peaks upregulated in *GATA6^{R456G/R456G}* cells were overlapped with *GATA6^{R456G/R456G}* ATAC-seq data to establish chromatin accessibility. Four percent of peaks enriched in *GATA6^{R456G/R456G}* cells were in open chromatin regions; these peaks were enriched for the CTCF and SP1 motifs.

The online version of this article includes the following figure supplement(s) for figure 5:

**Figure supplement 1.** Differential gene expression analysis of GATA6-bound genes.

At day 4, *GATA6* bound ~21,000 open chromatin peaks genome-wide, of which ~ 21% are near promoters. Notably, only 12% of *GATA6*-bound open chromatin peaks overlapped with ATAC peaks and were markedly enriched for a GATA-binding motif (*Figure 5A*). ChIP-seq data from day 8 cells were similar (*Supplementary file 4*). These data inferred that GATA6 directly participates in regulating transcriptional activation of *EOMES, GATA4, MEIS1/2, NKX2.5, TBX5*, and *TBX20* (*Supplementary file 4*).

By contrast, 88% of GATA-bound peaks occurred in closed chromatin (*Figure 5B*). Sequences within these peaks were not enriched for the GATA motif but were highly enriched for the MEF2 binding motif. MEF2 proteins contain a DNA-binding domain (MADS) and transcriptional activating domain (TAD) (*Backs and Olson, 2006*) that promote differentiation gene programs for cardiac, skeletal, and smooth muscle myocytes (*Potthoff and Olson, 2007*). During cardiomyocyte differentiation of WT hiPSCs MEF2 transcripts increased 120-fold between days 4–8 (*Supplementary file 2*). In comparison, at day 8–12, MEF2A and MEF2C levels were decreased in *GATA6^{+/-}* cells. As together, these data implied that GATA6 functioned as a cardiac pioneer factor, which has been proposed for other GATA proteins, we examined the correlation between GATA6 bound to closed chromatin at day 4 and gene expression at day 5 in WT cells. There were 2878 genes differentially expressed between day 4 and day 5 WT cells. Of those, 1049 were associated with GATA6-bound closed chromatin (36.4%). Among these, 583 (56%) genes had increased expression at day 5, including key cardiac developmental transcription factors, *GATA4, SMYD1, KDR,* and *TBX5* (*Figure 5—figure supplement 1*, *Supplementary file 5*). Based on these data, we propose that GATA6 is a cardiac development pioneer factor.

## Epigenetic abnormalities in *GATA6* mutant iPSCs

In comparison to WT, day 4 *GATA6^{+/-}* cells had~6500 diminished GATA6 ChIP-seq peaks and ~5800 diminished ATAC peaks (*Figure 6A* and *Figure 6—figure supplement 1*). Diminished ATAC peaks in proximity to promoters were enriched for GATA, HAND, HNF, and SOX motifs (*Figure 6—figure supplement 1*, *Figure 3*, *Figure 4* and *Supplementary file 2*), indicating that attenuated expression of these transcription factor family members could have direct functional consequences. Notably however, few diminished GATA-bound peaks resided within open chromatin (10.7%), of which only 35% were enriched for GATA motifs (*Figure 5—figure supplement 1B*, *Figure 6C*). Instead, and consistent with properties of a pioneer factor, *GATA6^{+/-}* cells had prominent loss (89%) and gain (97%) of GATA-bound peaks within closed chromatin (*Figure 6B–D*). From these data we infer that GATA interactions with closed chromatin are highly sensitive to protein dosage, while interactions with open chromatin at promoter regions remain relatively intact despite reduced GATA6 protein levels.

By contrast, day 4 *GATA6^{-/-}* cells contained~29,000 diminished ATAC-seq peaks of which ~ 31% resided in proximity to promoters (*Figure 6—figure supplement 1*). Profiles at day 8 cells were similar (*Figure 6—figure supplement 2*). Notably, diminished ATAC peaks showed enrichment in the binding motif for *NFY* that encodes the ubiquitous promoter element binding factor of CCAAT-boxes. These data implied that extinguishing *GATA6^{-/-}* expression caused widespread deficits in

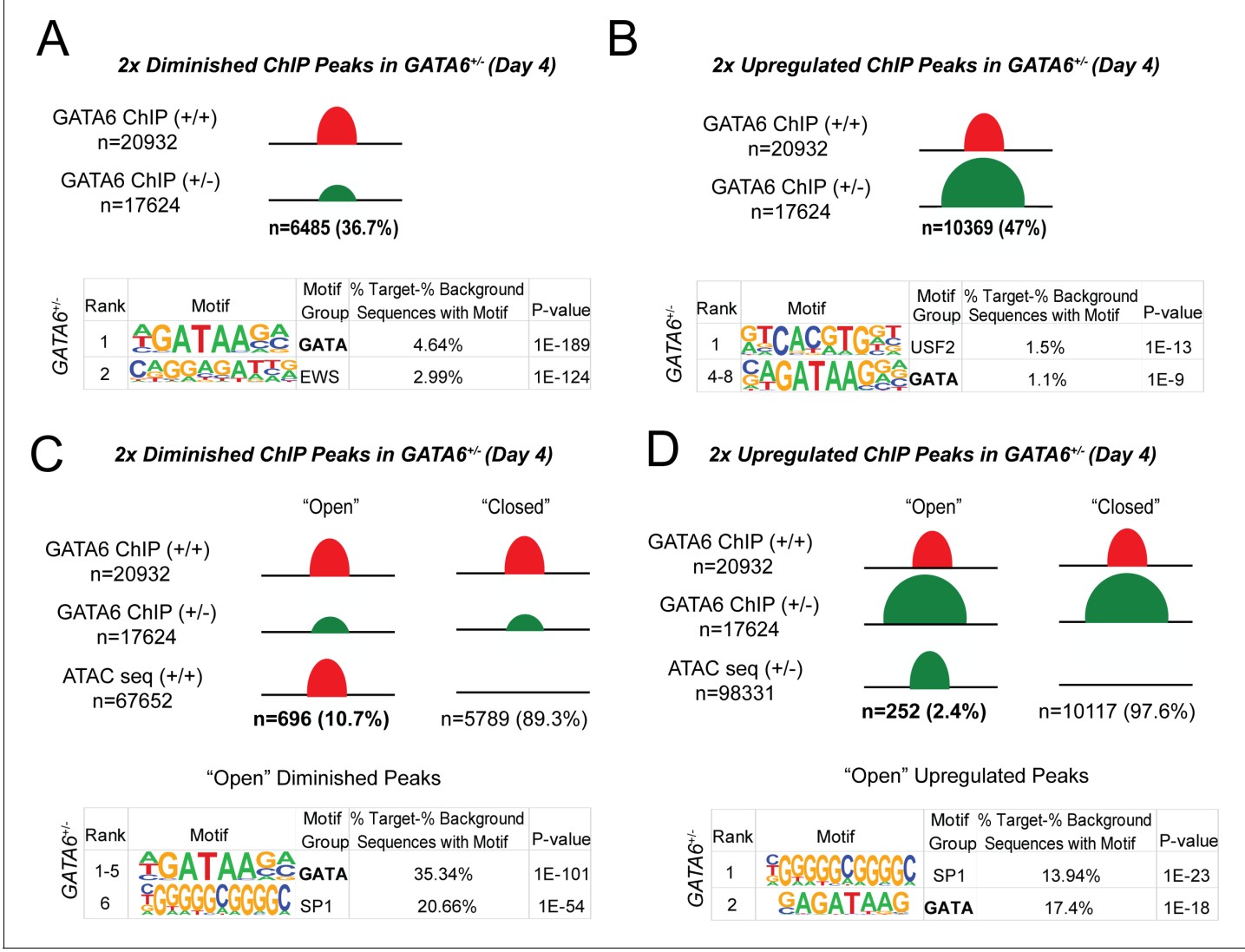

**Figure 6.** Epigenetic abnormalities in *GATA6*[+/-] cells. (**A**) Of the 20932 WT GATA6 ChIP-seq peaks, 39.3% were reduced in *GATA6*[+/-] cells (adjusted p<1e-4, two fold). These peaks were enriched for the GATA and EWS binding motifs by HOMER analysis. (**B**) Of the 17624 *GATA6*[+/-] ChIP-seq peaks, 47% were upregulated in *GATA6*[+/-] cells (adjusted p<1e-4, two fold). These peaks were enriched for the USF2 and GATA-binding motifs by HOMER analysis. (**C**) Peaks diminished in *GATA6*[+/-]cells were overlapped with WT ATAC-seq data to establish chromatin accessibility. Only 10.7% of peaks diminished in *GATA6*[+/-] cells were in open chromatin regions; these peaks were enriched for the GATA and SP1 motifs. (**F**) Peaks upregulated in *GATA6*[R456G/R456G] cells were overlapped with *GATA6*[+/-] ATAC-seq data to establish chromatin accessibility. Only 2.4% of peaks enriched in *GATA6*[+/-] cells were in open chromatin regions; these peaks were enriched for the SP1 and GATA motifs.

The online version of this article includes the following figure supplement(s) for figure 6:

**Figure supplement 1.** ATAC-seq analysis of cardiac genes in day 4 *GATA6* LoF and *GATA6*[R456G/R456G] hiPSC-CMs.

**Figure supplement 2.** ATAC-seq analysis of cardiac genes in day 8 *GATA6* LoF and *GATA6*[R456G/R456G] hiPSC-CMs.

promoter activation, which likely accounted for failed mesoderm specification and cardiomyocyte differentiation.

ChIP-seq analyses of *GATA6*[R456G/R456G] cells (*Figure 5C–F*, *Figure 5—figure supplement 1*, *Supplementary file 4*) identified ~8000 diminished peaks. In comparison to other mutant lines, *GATA6*[R456G/R456G] cells had a significantly higher proportion of diminished peaks (21%) in open chromatin that contained a GATA or HAND binding motif and 19% of these diminished peaks were associated with differential gene expression (*Figure 5—figure supplement 1C*). Additionally, missense cells had a remarkable number (~44,000) of augmented and ectopic peaks. We suggest that these

aberrant chromatin interactions reflected direct promiscuous GATA6$^{R456G}$ binding activity or perhaps occurred in response to aberrant RA signaling.

Among the upregulated peaks,~4% of these peaks had GATA-bound to open chromatin, with enrichment of the CTCF binding motif. Furthermore,~12% were associated with differential gene expression, including *MSX1, FOXA1, MEIS2, HCN4,* and *HOXA2* (*Supplementary file 4*). However, the vast majority of increased or ectopic peaks (96%) resided within closed chromatin and lacked the GATA6 binding motif. Even so, 14% of the differential closed peaks were associated with differential gene expression, including peaks in *TBX20, FOXA1,* and *FOXH1*. Our data suggest that the GATA6$^{R456G}$ variant impairs binding to the GATA motif, reducing normal GATA6 function, and also promotes promiscuous binding with either repressive or activating transcriptional effects.

## Phenotypes associated with epigenetic abnormalities in *GATA6* mutant iPSCs

We examined ATAC-seq and GATA6-ChIP-seq data to identify potential mechanisms for altered gene transcription with relevance to clinical phenotypes in CHD patients with pathogenic GATA6 variants. GATA6-bound peaks were universally decreased in *GATA6$^{+/-}$* cells, including those associated many cardiac developmental transcription factors genes (e.g., *HAND2, KDR,* and *TBX5 Figure 7A–C*) and these changes were associated with normal or modest reduction in transcript levels (*Supplementary file 2*). While many of these GATA6-bound peaks were also diminished in *GATA6$^{R456G/R456G}$* cells, the missense cells also had many enhanced and augmented peaks which were associated greater differential (decreased or increased) gene expression. For example, only *GATA6$^{R456G/R456G}$* cells had diminished GATA6-bound peaks identified in WT cells associated with *GATA4* (*Figure 7D*) and *MEF2A* and also new GATA6-bound peaks in *MEF2C, ZIC1* and *ZIC3* (*Figure 7E*). The distinct epigenetic profiles in *GATA6$^{R456G/R456G}$* may contribute to the greater dysregulation of cardiac gene expression and cardiomyocyte maturation in comparison to *GATA6$^{+/-}$* cells (*Figures 2* and *3* A-C). However regardless of whether epigenetic changes in *GATA6$^{R456G/R456G}$* cells decreased normal gene expression or erroneously activated the expression of other genes, the associated cardiac consequence of altered transcription was similar to that of *GATA6$^{+/-}$* cells – disruption of cardiac outflow tract development (*Figure 1*).

By contrast, altered GATA6 binding and chromatin accessibility with accompanying dysregulated transcripts likely contributed to the extra-cardiac phenotypes that disproportionately affect patients with heterozygous *GATA6* R456G and other exon four missense variants. Pancreatic development requires cooperative *GATA4* -*GATA6* interactions and compound *Gata4-null* and *Gata6*-haploinsufficient mice have reduced *Pdx1* expression, pancreatic agenesis or profound hypoplasia (*Carrasco et al., 2012*). In addition to reduced occupancy of GATA6-motifs in *GATA4* with striking attenuated expression (*Figure 7D*, *Supplementary file 2*, *Supplementary file 4*), GATA6 R456G also aberrantly bound chromatin associated with *FOXA2, DEANR1* (*LINC00261*), and *PDX1* and diminished transcription of these genes (*Figure 7—figure supplement 1*). *DEANR1* encodes a long noncoding RNA that specifies endoderm and pancreatic lineages by regulating the expression of *FOXA2* (*Jiang et al., 2015*) which in turn regulates *PDX1* expression (*Gao et al., 2008*).

Aberrant RA signaling may have contributed to the abnormal epigenetic profiles in *GATA6$^{R456G/R456G}$* cells. In missense cells, prominent ATAC peaks were identified in association with binding motifs for GATA, SOX6, HOX, and RARa in the promoter and exon 1 of *ALDH1A2* and transcript levels were increased. *SOX6* also had augmented ATAC peaks with RARa and GATA motifs, and GATA6$^{R456G}$ bound an intron one motif (*Figure 7—figure supplement 2A–B*). *GATA6$^{R456G/R456G}$* cells also showed increased ATAC peak heights with RARa binding motifs in close proximity to the reported RA-responsive element in *HOXB1* that promotes foregut expression. We also observe GATA6$^{R456G}$ protein aberrantly bound to the *STRA6* locus, potentially affecting expression of this gene (*Figure 7—figure supplement 2C*) These data suggest multiple direct and indirect epigenetic mechanisms by which the GATA6$^{R456G}$ variant distinctly perturbed gene programs and resulted in maldevelopment of the diaphragm, pancreas and other abdominal organs in addition to cause cardiovascular malformations.

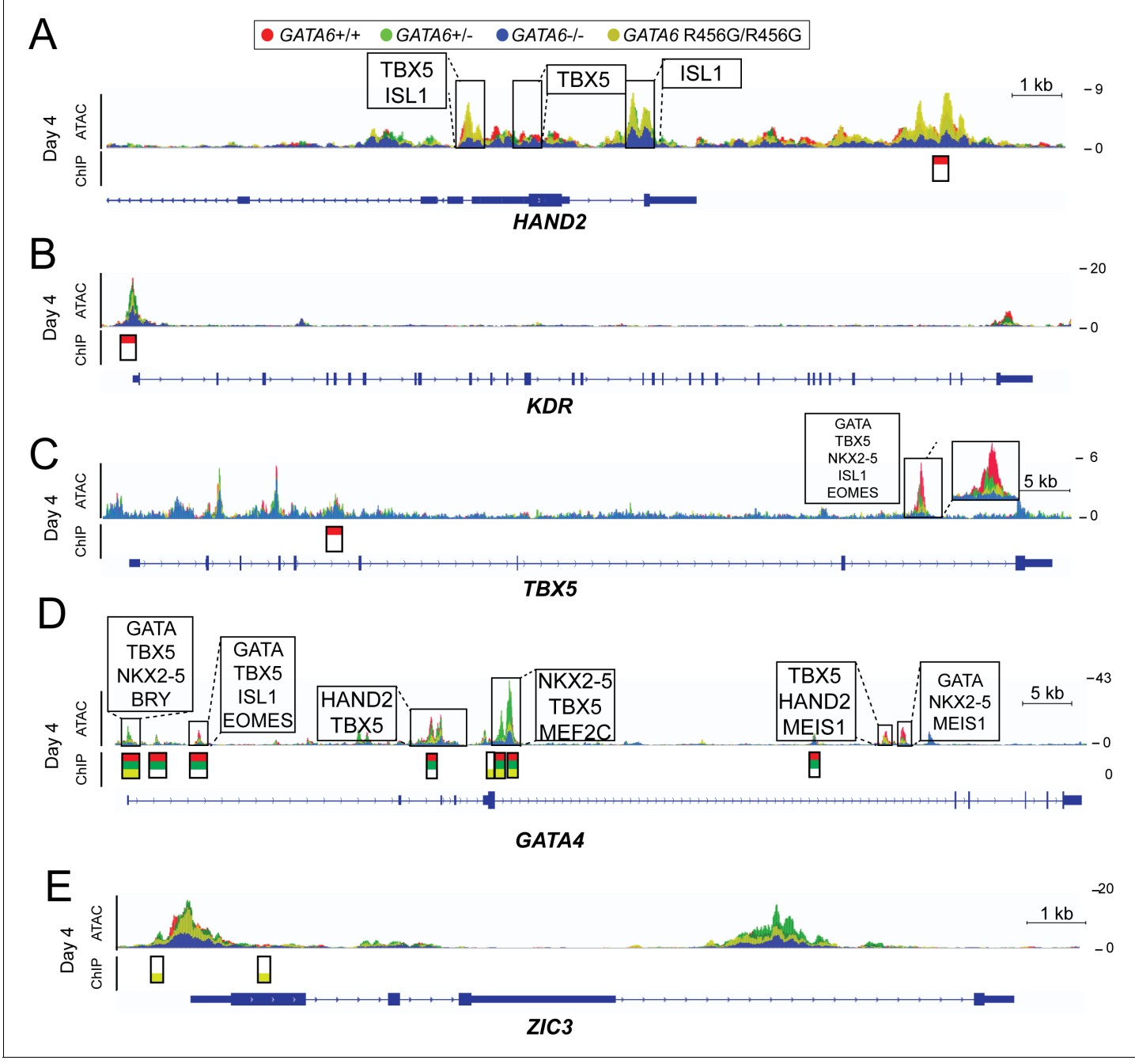

**Figure 7.** ATAC-seq and GATA6 ChIP-seq analysis of *GATA6* variant hiPSC-CMs reveals aberrant binding to congenital heart disease genes. ATAC peaks (upper), GATA6 ChIP peaks (lower) and DNA-binding motifs (upper, boxed) found in day 4 *GATA6* LoF and *GATA6*[R456G/R456G] cells, visualized using the Integrative Genomics Viewer (IGV) (A) Lost GATA6 ChIP-seq peak in the *HAND2* locus, with differential chromatin accessibility (ATAC-seq). (B) Lost GATA6 ChIP-seq peak at the *KDR* promoter, leading to reduced chromatin accessibility. (C) Lost GATA6 ChIP-seq peak in *TBX5*. (D) *GATA6*[R456G] does not bind the *GATA4* locus in regions of open chromatin. (E) *GATA6*[R456G] ectopically binds the *ZIC3* promoter.

The online version of this article includes the following figure supplement(s) for figure 7:

**Figure supplement 1.** ATAC-seq and GATA6 ChIP-seq analysis of *GATA6* variant hiPSC-CMs reveals misregulation of pancreatic genes.

**Figure supplement 2.** ATAC-seq and ChIP-seq of *GATA6* variant hiPSC-CMs reveals altered expression and chromatin accessibility in retinoic acid signaling-related genes.

## Discussion

By combining the strengths of WES, CRISPR/Cas9 genome editing, and hiPSCs, we demonstrate molecular and developmental mechanisms by which damaging variants in *GATA6* cause cardiac and extra-cardiac congenital anomalies. Transcriptional and epigenetic analyses of differentiating cardiomyocytes from isogenic WT and mutant hiPSCs provide evidence that GATA6 functions as a pioneer factor that modifies chromatin accessibility and promotes the expression of gene networks involving HAND2, VEGFR and neural crest cells that enable second heart field development and patterning of the outflow tract and atria. Our data also supports critical roles for *GATA6* in endodermal and retinoic acid signaling gene networks that gives rise to the pancreas and diaphragm. We find that LoF *GATA6* variants repress epigenetic modification and transcriptional activity, while an exon 4 *GATA6* missense variant alters normal and also causes ectopic epigenetic effects that enhanced retinoic acid signaling resulting in profound deleterious consequences on gene expression. The selective transcriptional deficits evoked by these variants provides mechanistic insights into the high prevalence of particular heart malformation and associated pancreatic dysgenesis and congenital diaphragmatic hernias that occur in CHD patients with pathogenic *GATA6* variants (*Figure 8*).

Our analyses demonstrate that some *GATA6* expression is required for lineage specification of human cardiomyocytes. When cultured to promote cardiomyocyte differentiation *GATA6^-/-* cells had markedly reduced promoter activation with *NFY* transcripts that encode the binding factor of CCAAT-box, a nearly ubiquitous promoter element. *GATA6^-/-* cells were viable but showed very limited transcriptional evidence of mesoderm specification and extinguished cardiomyocyte gene expression. Instead *GATA6^-/-* cells adopted a gene program suggestive of nonspecific cells with expression of neuroectodermal and fibroblast genes.

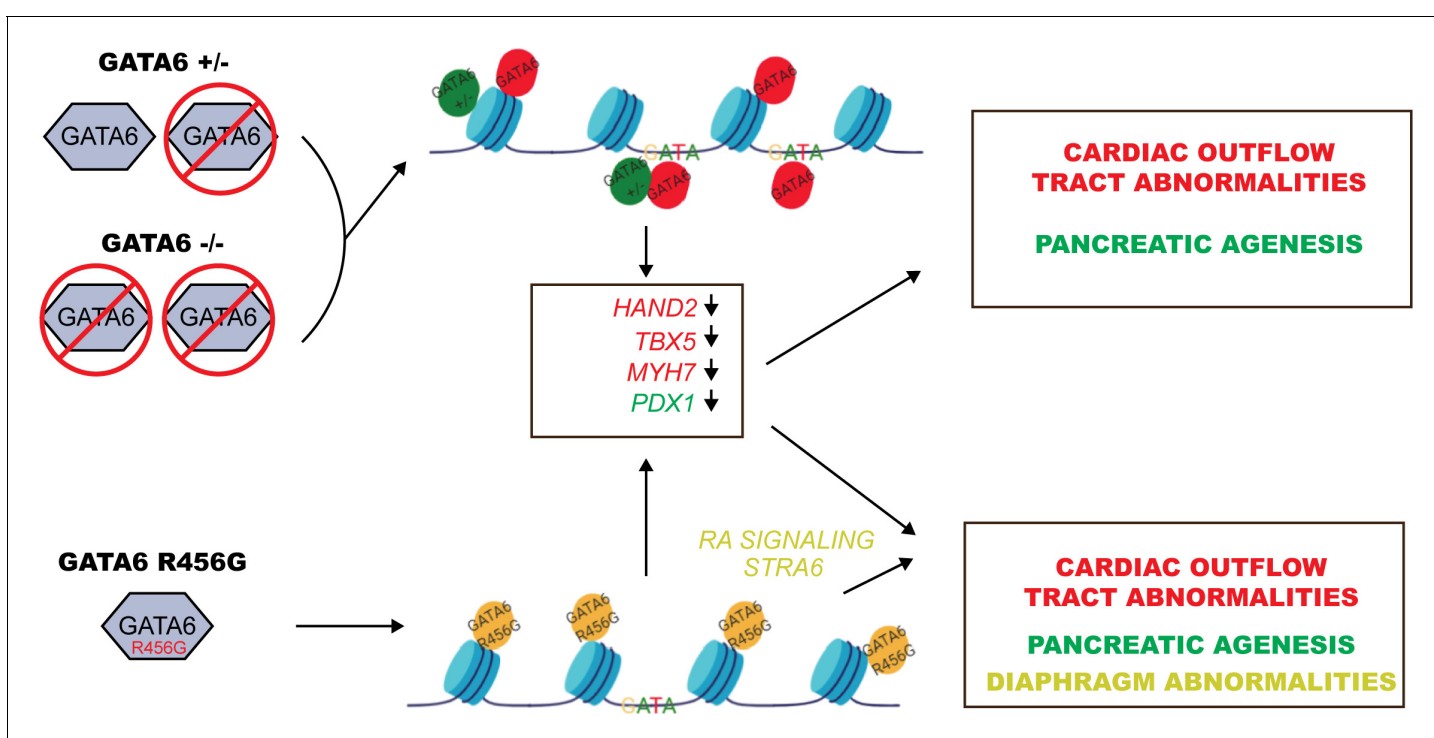

**Figure 8.** Model for GATA6 transcriptional regulation of cardiac and pancreatic gene expression (see Discussion). ChIP-seq and ATAC-seq data of WT hiPSCs identified GATA6 bound to closed chromatin in intergenic regions without a GATA-binding motif. Moreover, GATA6 binding was associated with temporal activation of transcription in nearby genes that activate cardiomyocyte and endoderm gene network. These findings indicate that *GATA6* engages chromatin and fosters a competent state for transcription factor binding and transcriptional activation, supporting the conclusion that GATA6 is a pioneer factor, as is suggested for other GATA proteins (*Fisher et al., 2017*). Notably, nonsense-mediated decay of *GATA6^+/-* transcripts reduced GATA6 protein levels, altered chromatin accessibility and decreased gene transcription, implying that intergenic sites are sensitive to GATA6 dosage. In addition, GATA6 functions as a traditional transcription factor, binding GATA motifs in promoters and activating transcription. These functions were relatively insensitive to half normal GATA6 levels.

These observations are consistent with prior analyses of *Gata6* ablation in mice that exhibit early embryonic lethality due to defects in extraembryonic endoderm formation (*Gottlieb et al., 2002*; *Morrisey et al., 1998*; *Zhao et al., 2005*). While a tetraploid complementation system reported circumvention of this defect (*Zhao et al., 2005*), only diminutive *Gata6*-null embryos were recovered with poor tissue integrity and embryonic resorption after E10.5. Although reverse-transcription-PCR of 'rescued' E9.5 *Gata6*-null embryos identified transcripts associated with early cardiomyocyte lineages, similar transcripts were absent in $GATA6^{-/-}$ hiPSCs from which no cardiomyocytes emerged. Perhaps these differences reflect diffusible factors from other cell lineages not present in hiPSCs cultured in vitro, species-specific dosage requirements, or other mechanisms.

*GATA6* haploinsufficiency influenced cardiomyocyte differentiation. ATAC and GATA6 ChIP-seq peaks were depleted in multiple developmental cardiac genes including *SMYD1*, *NKX2-5*, and *ISL1* in $GATA6^{+/-}$ lines. As *SMYD1* regulates *HAND2* expression (*Gottlieb et al., 2002*), depression of *SMYD1* transcripts likely resulted in a cascade effect, depleting *HAND2*-network genes. Notably, transcriptional deficits did not persist at differentiation day 30 $GATA6^{+/-}$ hiPSC-CMs when many cardiac transcription factors (*SMYD1*, *HAND2*, *NKX2-5*, *GATA4*, and *MEIS1)* were expressed at levels found in WT cells. We presume that a compensatory transcriptional mechanism restored sufficient expression of many gene programs. However, *ISL1* transcripts remained abnormally high, and expression profiles of fetal and adult myosins (high *MYH6* and low *MYH7*, respectively) indicated immaturity of $GATA6^{+/-}$ hiPSC-CMs. Persistence of this developmental deficit in vivo could contribute to longitudinal adverse cardiac outcomes in patients with pathogenic *GATA6* variants.

Our cardiomyocyte differentiation protocol also activated endodermal gene programs and uncovered misexpression in mutant lines. Aberrant but variable epigenetic profiles were identified in $GATA6^{+/-}$ cells and in $GATA6^{R456G/R456G}$ cells that led to misexpression of *HNF* and *FOXA* gene family-members occurred in both mutant lines. For example, both mutant lines had diminished GATA6 binding of open chromatin binding to *HNF1* with reduced transcript levels. By contrast, only $GATA6^{R456G/R456G}$ cells lacked GATA6 binding to open chromatin near *DEANR1* a regulator of *FOXA2* expression (*Gao et al., 2008*). $GATA6^{R456G/R456G}$ cell also reduced expression of *GATA4*. While these data illustrate multiple mechanisms for depleting *PDX1* expression and account for the prominent association of CHD and pancreatic agenesis in patients with *GATA6* LoF and exon four missense variants, the pathways leading to this shared consequence were strikingly different. Epigenetic changes observed in $GATA6^{+/-}$ cells depleted normal GATA6 binding to chromatin with attenuated gene transcription. Epigenetic changes observed in $GATA6^{R456G/R456G}$ cells were far more sweeping, repressing many more genes and erroneously activating others.

We found that $GATA6^{R456G/R456G}$ cells strikingly increased activation of RA signaling. Indeed, early inhibition of excessive RA signaling normalized some of the many aberrant in $GATA6^{R456G/R456G}$ cells. Transcriptional data showed increased expression of the RA biosynthetic enzyme encoded by *ALDH1A2* and of *STRA6*, which encodes the integral membrane receptor that triggers release and uptake of retinol from circulating retinol-binding protein (*Chen et al., 2016*). As RARa motifs occur in ATAC peaks associated with *HOXA2*, *HOXB1,* and *SOX6* genes and others, we deduced that $GATA6^{R456G/R456G}$ likely activates transcription of these genes by increasing RA signaling. Moreover, as ATAC peaks in the *ALDH2A2* contain SOX6-binding motifs, enhanced *SOX6* expression would increase RA biosynthesis that in turn would further increase *STRA6* expression (*Bouillet et al., 1997*) to further amplify RA signaling.

Human congenital anomalies arise from inadequate and excessive expression of key developmental signaling genes. Our analyses support the conclusion that CHD, pancreatic malformations and congenital diaphragmatic hernia that occur in *GATA6* exon four missense mutants reflect both loss of physiologic levels of GATA6 and excessively activated RA signaling. Development of the second heart field and neural crest cell migration into the nascent outflow tract is regulated by RA signaling (*Zaffran and Kelly, 2012*), and when inappropriate, cardiac morphogenesis is perturbed (*Perl and Waxman, 2019*). RA signaling also regulates transcriptional signals to develop the diaphragm (*Kardon et al., 2017*), particularly formation of the central tendon (*Coles and Ackerman, 2013*) around which diaphragmatic myocytes are patterned. As RA signals inhibit the expression of myogenic specification genes, including *PAX3* (*El Haddad et al., 2017*; *Magli et al., 2019*), excessive RA signaling could impair diaphragm formation by impairing tendon development in addition to altering differentiation and maturation of diaphragmatic myocytes. Increased RA signaling also augments *SOX6* expression, which attenuates *PDX1* activation of insulin signaling. When combined with

deficient expression in critical endodermal genes (*FOXa1 FOXa2*), specification and differentiation of pancreatic progenitors are likely to be impaired. Consistent with these RA-mediated mechanisms, we note that human mutations in *STRA6* cause syndromic congenital diaphragmatic hernia with both CHD and pancreatic anomalies (*Golzio et al., 2007*; *Pasutto et al., 2007*).

Based on the heightened risk for congenital diaphragmatic hernia and pancreatic agenesis only among patients with missense variants within exon 4, we suggest that the encoded ZF domain in GATA6 has critical interactions with chromatin and DNA sequences that evoke the epigenetic and transcriptional consequences. Exon four missense residues perturb these functions- culminating in augmented RA signaling and extinguished *PDX1* expression. We suggest that missense variants residing outside of exon four do participate in these molecular processes, nor convey similar risks for developmental anomalies. Conformation of this hypothesis would improve clinical interpretation of *GATA6* genotypes.

Our studies also demonstrate that in vitro differentiation of hiPSCs evoke developmental gene expression profiles, independent of the many three-dimensional morphological cues that occur in vivo. This system has the potential to illuminate the consequences of lethal mutations that difficult to study in model organisms. While we recognize the limitations of cell models, by interpreting our molecular data with prior mouse studies and clinical studies of human patients, we believe that these iPSC studies provide valid molecular mechanism by which human *GATA6* mutations cause cardiac outflow tract defects, pancreatic agenesis, and congenital diaphragmatic hernias.

In conclusion, we show that genetic perturbation of a pioneer factor, GATA6, altered network-level transcriptional pathways that are critically involved in development of the heart (*GATA4*, *HAND2*), endodermal lineages (*HNF*, *FOXA1*, *FOXA2*), and pancreas (*PDX1*, *SOX6*) and diaphragm (*NR2F2*, *STRA6*, *ZFPM2*). Development of these organs is highly sensitive to changes in *GATA6* gene dosage from LoF variants and from missense variants in the second zinc-finger domain that alter RA signaling.

The combination of WES, genome sequencing data, CRISPR/Cas9 genome editing, and molecular analyses of developmentally immature human iPSC-derived lineages represents a valuable paradigm for mechanistic studies of human development. We expect that this platform will continue to illuminate molecular understandings of organogenesis that may improve clinical use of genetic data for personalized medicine.

## Materials and methods

**Key resources table**

| Reagent type (species) or resource | Designation | Source or reference | Identifiers | Additional information |
|---|---|---|---|---|
| Cell line (human) | PGP1 | *Lee et al., 2009* | GM23338 | Male; mycoplasma-free |
| Cell line (human) | TNNT2-GFP | This paper | | Derived from PGP1 cell line GM23338, mycoplasma-free |
| Commercial assay or kit | Zero Blunt TOPO PCR Cloning Kit | ThermoFisher | K280002 | |
| Commercial assay or kit | Human Stem Cell Nucleofector Kit | Lonza | VPH-5022 | |
| Commercial assay or kit | Nextera XT Sample Preparation Kit | Illumina | FC-131–1096 | |
| Commercial assay or kit | Nextera DNA Sample Kit (ATAC-seq) | Illumina | FC-121–1030 | |
| Commercial assay or kit | Tru ChIP Chromatin Shearing Hit | Covaris | 520154 | |
| Commercial assay or kit | Chromium i7 Multiplex Kit | 10X Genomics | 1000073 | |
| Commercial assay or kit | Chromium Chip B Single-Cell Kit | 10X Genomics | 1000075 | |

*Continued on next page*

*Continued*

| Reagent type (species) or resource | Designation | Source or reference | Identifiers | Additional information |
|---|---|---|---|---|
| Sequence-based reagent | Guide RNAs | This paper | | *GATA6* Exon 2 Guide 1: GAGCCCCTACTCGCCCTACG *GATA6* Exon 2 Guide 2: GCCCCTACTCGCCCTACGTG *GATA6* Exon 4 Guide 1: GGCGTTTCTGCGCCATAAGG |
| Sequence-based reagent | GATA6 sequencing primers | This paper | PCR Primers | GATA6 Exon 2 Sequencing Primer Left/Forward GACGTACCACCACCACCA GATA6 Exon 2 Sequencing Primer Right/Reverse CTTACCTGCACTGGGACCC GATA6 Exon 4 Sequencing Primer Left/Forward TGAATTCACGGAGACAGGCT GATA6 Exon 4 Sequencing Primer Right/Reverse TACAAGTGAGCAGAATACATGGCA |
| Sequence-based reagent | ATAC-seq amplification oligos | *Buenrostro et al., 2015* | | |
| Recombinant DNA reagent | Cas9 plasmid | Addgene | PX459v2 | |
| Chemical compound, drug | WIN 18446 | Tocris | | |
| Antibody | rabbit mono-clonal Gata6 | Cell Signaling Technology | 5851S | 10 ug/ChIP |
| Antibody | rabbit mono-clonal Gata6 | Abcam | Ab175927 | 1:1000 dilution |
| Software, algorithm | RNA-seq pipeline: bcbio-nextgen | *Chapman et al., 2020* | v.1.2.3 | Hg19 |
| Software, algorithm | R Package: DESEQ2 | *Love et al., 2014* | v. 2.1.18.1 | |
| Software, algorithm | R Package: Seurat | *Stuart et al., 2019* | v.3.0.0 | |
| Software, algorithm | R Package: ChIP-seeker | *Yu et al., 2015* | v.1.14.1 | |
| Software, algorithm | HOMER | *Heinz et al., 2010* | v4.10.3 | |

## Human subjects

CHD subjects were recruited to the Congenital Heart Disease Network Study of the Pediatric Cardiac Genomics Consortium (CHD GENES: ClinicalTrials.gov identifier NCT01196182) after approval from Institutional Review Boards as previously described (*Jin et al., 2017*; *Gelb et al., 2013*). Written informed consent was received from subjects or their parents prior to inclusion in the study. Any subject with CHD, regardless of age, race, or ethnicity was eligible for enrollment. Subjects with variants in GATA6 and 2nd heart field genes including CHD diagnoses, and extra-cardiac phenotypes are provided in *Supplementary file 1*.

## Exome sequencing and analyses

DNA was extracted peripheral blood samples and captured using the Nimblegen v.2 exome capture reagent (Roche) or Nimblegen SeqxCap EZ MedExome Target Enrichment Kit (Roche) followed by Illumina DNA sequencing as previously described (*Jin et al., 2017*). Reads were mapped to the reference genome (hg19), and further processed using the GATK Best Practices workflows as previously described (*Jin et al., 2017*; *McKenna et al., 2010*). Single nucleotide variants (SNVs) and small indels were called with GATK HaplotypeCaller and filtered for rarity (ExAC allele frequency $\leq 10–5$) and quality as previously described (*Jin et al., 2017*). The MetaSVM algorithm was used to predict deleteriousness of de novo missense mutations (annotated as 'D-Mis') using software defaults (*Dong et al., 2015*; *Sulahian et al., 2014*).

## TNNT2-GFP, *GATA6$^{+/-}$*, *GATA6$^{-/-}$*, and *GATA6$^{R456G/R456G}$* hiPSCs and iPSC-CMs

All hiPSCs and hiPSC-CMs used here are derived from the male parent iPSC line PGP1 (*Akerman et al., 2017*; RRID-GM23338) and are mycoplasma-free. The TNNT2-GFP iPSC line was generated by homology-directed repair, using a plasmid with the endogenous TNNT2 sequence connected to a GSSSS linker region, attached to the GFP gene, and the entire construct was flanked by 500 bp homology arms. PGP1 iPSCs were transfected as described (*Sharma et al., 2018b*) to obtain homozygous integration of the GFP tag in both TNNT2 alleles. Edited clones were selected using puromycin and expanded and genotyped as previously described (*Sharma et al., 2018b*) and differentiated into cardiomyocytes as described (*Sharma et al., 2018a*).

*GATA6$^{+/-}$* and *GATA6$^{-/-}$* hiPSCs were generated from TNNT2-GFP and PGP1 iPSCs by non-homologous end joining using a 2 µg plasmid expressing Cas9 (PX459v2 from Addgene) that was co-transfected with 2 µg plasmid expressing guide RNA (provided in Key Resource Table) using a stem cell nucleofector kit (Amaxa). Edited clones were selected using puromycin and expanded and genotyped as previously described (*Sharma et al., 2018b*) and differentiated into cardiomyocytes as described (*Sharma et al., 2018a*).

*GATA6$^{R456G/R456G}$* hiPSCs were generated by homology-directed repair using a 2 µg plasmid expressing Cas9 (PX459v2 from Addgene) that was co-transfected with 2 µg plasmid expressing guide RNA (provided in Key Resource Table) and 2 µg single-stranded oligonucleotide (HDR template) using a stem cell nucleofector kit (Amaxa). Edited clones were selected using puromycin and expanded and genotyped as previously described (*Sharma et al., 2018b*) and differentiated into cardiomyocytes as described (*Sharma et al., 2018a*).

## Confirmation of editing in subcloned iPSC lines

All edited lines sub-cloned, by dissociating 1000 genome-edited hiPSCs cells, filtering through a 60 µm strainer, and evenly distributing them onto a 15 cm dish containing Matrigel and mTeSR+ rho kinase inhibitor. Individual monoclonal hiPSC colonies were picked when colony size reached approximately 200 cells and placed into individual separate wells of a 96 well plate. Clones were allowed to grow to 80% confluency, at which time a sample was obtained for PCR amplification to verify the *GATA6* variant or GFP-tagged *TNNT2* and to assess zygosity. PCR-amplified fragments (primers provided in Key Resource Table) containing putative variants were submitted for Sanger sequencing and next-generation sequencing. Sanger sequencing traces were deconvoluted using TIDE software to confirm zygosity. Clones carrying *GATA6* mutations were further expanded for subsequent culture and differentiation.

## hiPSC-CM differentiation

The hiPSC-CMs were generated from *GATA6* mutant hiPSCs using a small-molecule mediated differentiation approach that modulates Wnt signaling (*Sharma et al., 2018a*). Cells began beating at approximately day 7 post-differentiation. Cardiomyocytes were metabolically selected from other differentiated cells by using glucose deprivation as previously described (*Sharma et al., 2018a*). hiPSC-CM Western blots: GATA6 protein expression and nuclear localization were performed using ab175927.

## RNA-sequencing and analysis

RNA-seq experiments were performed on at least two independent cultured cell samples per time point, with the exception of *GATA6$^{R256G/R256G}$* DMSO control in the RA-signaling experiment. Trizol (Thermo Fisher) was used to harvest differentiating *GATA6* mutant hiPSC-CMs (days 0, 4, 8, 12, 30 of differentiation) designated for RNA-sequencing analysis. Samples harvested in Trizol were stored at −80˚C until RNA was extracted. All RNA samples had an RNA integrity number of ≥8. Library preparation was conducted using the Nextera library preparation method. RNA-Seq library samples were pooled and run on four lanes (one flow cell) using the Illumina NextSeq500 platform. All data was combined into a single fastq file. Samples typically had 30–50M reads each. The raw reads were aligned by HISAT2 (v.2.1.0) to human genome (hg38). The aligned reads were quantified by FeatureCounts, counts were normalized, and differentially expressed genes were identified using DESeq2 (v1.24.0). DESeq2 data was analyzed and visualized in R using the ggplot2 (v3.1.0), pheatmap

(v1.0.12), gProfiler (v0.6.7), and VennDiagram (v1.6.20) packages. Single-cell RNA-seq analysis for hiPSC-CMs was conducted using a 10x Genomics Chromium platform and analyzed in R using the Seurat (v. 3.5.1) pipeline. Seurat was used to regress out cell-cell variation in cell complexity, gene expression driven by batch, cell alignment rate, the number of detected molecules, and mitochondrial gene expression. Starting with Seurat v2.0, Seurat implements this regression as part of the data scaling process.

## ATAC-seq and Hi-C chromatin analysis

ATAC-seq was performed as previously described (*Buenrostro et al., 2015*; *Corces et al., 2017*). Briefly, 50,000 cells were harvested and lysed to isolate nuclei. Nuclei were treated with Tn5 transposase (Nextera DNA Sample Prep Kit, Illumina) and DNA was isolated. Fragmented DNA was amplified using bar-coded PCR primers (defined in *Buenrostro et al., 2015*) and libraries were pooled. Pooled libraries were sequenced (Illumina Next-Seq) to a depth of 100 million reads per sample. Reads were aligned to the hg19 reference genome using BWA-MEM and peaks were called using HOMER (v4.10.3) (http://homer.ucsd.edu/homer/index.html). Functional analysis of ATAC-seq peaks was performed using ChIP-Seeker (v.1.14.1). Motif enrichment was performed using HOMER (v4.10.3). Differential peaks were called using HOMER (v4.10.3).

## ChIP-seq

Cells (6-well plates) were grown to approximately 80% confluency and then fixed with 1% fresh formaldehyde diluted in media. Cells were quenched with glycine, washed, and harvested. Cell pellets were stored at −80°C. Cells were thawed and nuclei were prepared using the Covaris Tru ChIP Chromatin Shearing Kit (Covaris, MA, USA). Chromatin was sheared using the Covaris E210 to an average fragment size of 200–700 bp and an input sample was purified. Chromatin (5 µg) was incubated with GATA6 antibody (RRID:AB_5851S) at 4°C O/N and then added to Protein G beads for 2 hr at 4°C. Beads were washed and bound chromatin was eluted at 65°C for 30 min on a heated vortex. DNA was then purified and quantified. ChIP-seq libraries were prepared using Nextera XT DNA sample prep (Illumina). Sequences were aligned to hg19 using BWA-MEM. Peaks were called using MACS2 using default parameters with a q value of 0.001. Differential peaks were called using HOMERv4.10.3.

## Retinoic acid inhibitor treatment assay

hiPSCs were seeded and differentiated to day 4, and then treated for 24 hr with either DMSO or 1 µM WIN 18446 (Tocris), an inhibitor of the ALDH1A2 enzyme. Cells were harvested and processed for RNA-seq.

## Statistical analyses

The distribution of damaging human variants across the *GATA6* gene was analyzed using the binominal test, implemented in R. The association of $GATA6^{+/-}$ and $GATA6^{R456G/+}$ variants and pancreatic agenesis or congenital diaphragmatic hernia was analyzed using the Fisher Exact test, implemented in R. Transcriptional responses of iPSC-CMs with or without a *GATA6* variant was compared using the Student's t-test was used for comparison. All error bars refer to standard deviation, unless otherwise specified. A p value of $< 0.05$ was considered significant.

## Acknowledgements

We gratefully acknowledge the NHLBI Pediatric Cardiac Genomics Consortium (PCGC) and Cardiovascular Development Consortium (CVDC) investigators for their support and expertise regarding the mechanisms driving congenital heart disease. Funding support for this study was provided in part by the NHLBI CVDC (U01HL098166) PCGC grants (U01-HL098188, U01-HL098147, U01-HL098153, U01-HL098163, U01-HL098123 and U01-HL098162), Fondation Leducq, the Engineering Research Centers Program of the National Science Foundation (NSF Cooperative Agreement No. EEC-1647837), NIH T32 HL116273, the Wellcome Trust (Sir Henry Wellcome fellowship), the German Academic Scholarship Foundation, and the Howard Hughes Medical Institute.

## Additional information

### Funding

| Funder | Grant reference number | Author |
| --- | --- | --- |
| National Institutes of Health | UM1HL128711 | George Porter<br>Martin Tristani-Firouzi<br>Deepak Srivastava<br>Jonathan G Seidman<br>Christine E Seidman |
| National Institutes of Health | UM1HL128761 | Christine E Seidman |
| National Institutes of Health | UM1HL098147 | Daniel M DeLaughter |
| National Institutes of Health | U01-HL098153 | Jonathan G Seidman<br>Christine E Seidman |
| National Institutes of Health | U01-HL098163 | Jonathan G Seidman<br>Christine E Seidman |
| National Institutes of Health | U01-HL098123 | Jonathan G Seidman<br>Christine E Seidman |
| National Institutes of Health | U01-HL098162 | Jonathan G Seidman<br>Christine E Seidman |
| National Science Foundation | EEC-1647837 | Jonathan G Seidman<br>Christine E Seidman |
| National Institutes of Health | T32HL116273 | Arun Sharma |
| Howard Hughes Medical Institute | | Lauren K Wasson |

The funders had no role in study design, data collection and interpretation, or the decision to submit the work for publication.

### Author contributions

Arun Sharma, Conceptualization, Resources, Data curation, Formal analysis, Supervision, Funding acquisition, Investigation; Lauren K Wasson, Conceptualization, Data curation, Formal analysis, Investigation; Jon AL Willcox, Sarah U Morton, Joshua M Gorham, Daniel M DeLaughter, Meraj Neyazi, Manuel Schmid, Data curation, Investigation; Radhika Agarwal, Min Young Jang, Christopher N Toepfer, Tarsha Ward, Yuri Kim, Alexandre C Pereira, Steven R DePalma, Angela Tai, Seongwon Kim, David Conner, Daniel Bernstein, Bruce D Gelb, Wendy K Chung, Elizabeth Goldmuntz, George Porter, Martin Tristani-Firouzi, Deepak Srivastava, Investigation; Jonathan G Seidman, Conceptualization, Funding acquisition, Investigation; Christine E Seidman, Conceptualization, Resources, Formal analysis, Supervision, Funding acquisition, Investigation; Pediatric Cardiac Genomics Consortium, Resources

### Author ORCIDs

Arun Sharma (iD) https://orcid.org/0000-0002-0607-5455
Lauren K Wasson (iD) https://orcid.org/0000-0002-5193-5215
Elizabeth Goldmuntz (iD) http://orcid.org/0000-0003-2936-4396
Deepak Srivastava (iD) http://orcid.org/0000-0002-3480-5953
Jonathan G Seidman (iD) http://orcid.org/0000-0002-9082-3566
Christine E Seidman (iD) https://orcid.org/0000-0001-6380-1209

### Ethics

Human subjects: CHD subjects were recruited to the Congenital Heart Disease Network Study of the Pediatric Cardiac Genomics Consortium (CHD GENES: ClinicalTrials.gov identifier NCT01196182) after approval from Institutional Review Boards as previously described (Pediatric Cardiac Genomics et al., 2013; Jin et al., 2017). Written informed consent was received from subjects or their parents prior to inclusion in the study. Clinical diagnoses were standardized based on review of medical data and family interviews.

Decision letter and Author response
Decision letter https://doi.org/10.7554/eLife.53278.sa1
Author response https://doi.org/10.7554/eLife.53278.sa2

## Additional files

### Supplementary files

• Supplementary file 1. Damaging Variants in *GATA6* and Second Heart Field Genes Associated with CHD and Extra-Cardiac Phenotypes.

• Supplementary file 2. Differential Genes Expressed in *GATA6* Mutant iPSCs and iPSC-CMs.

• Supplementary file 3. RNA-Seq-based principal component analysis (PCA) for GATA6 LoF and R456G missense hiPSC-CMs during differentiation shows an enrichment for cardiac genes.

• Supplementary file 4. GATA6 ChIP-seq peaks in WT and GATA6 Mutant iPSCs and iPSC-CMs.

• Supplementary file 5. Differential Genes Expressed in WT iPSC-CMs between day 4 and day 5 of differentiation.

• Supplementary file 6. Diminished ATAC peaks (≥2 fold) in day 4 GATA6 LoF and missense cells.

• Supplementary file 7. DNA-binding motifs found in ≥2 fold diminished ATAC peaks in gene promoters (0–10 kb from TSS) of differentially expressed genes in day 4 GATA6 LoF and missense cells.

• Transparent reporting form

### Data availability

All data generated or analyzed during this study are included in the manuscript.

The following previously published datasets were used:

| Author(s) | Year | Dataset title | Dataset URL | Database and Identifier |
|---|---|---|---|---|
| Verzi MP | 2010 | GATA6 ChIP-seq in differentiated cells | https://www.ncbi.nlm.nih.gov/geo/query/acc.cgi?acc=GSM575227 | NCBI Gene Expression Omnibus, GSM575227 |
| de Santa Pau EC | 2015 | GATA6 ChIP-seq | https://www.ncbi.nlm.nih.gov/geo/query/acc.cgi?acc=GSM1151694 | NCBI Gene Expression Omnibus, GSM1151694 |
| Shivdasani R | 2013 | Genome-wide analyses of GATA6 occupancy and functions provide insights into its oncogenic mechanisms in human gastric cancer | https://www.ncbi.nlm.nih.gov/geo/query/acc.cgi?acc=GSE51936 | NCBI Gene Expression Omnibus, GSE51936 |
| Verzi MP | 2010 | GATA6 ChIP-seq in proliferating cells | https://www.ncbi.nlm.nih.gov/geo/query/acc.cgi?acc=GSM575226 | NCBI Gene Expression Omnibus, GSM575226 |

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
