## [Decision Letter]

**Acceptance summary:**

Your study clearly defines new roles for *GATA6* in the development of multiple organs. Your study also provides new molecular targets of *GATA6*, generating possible therapeutics targets for cardiac, pancreatic, and diaphragmatic congenital diseases.

**Decision letter after peer review:**

Thank you for submitting your article "*GATA6* mutations in induced pluripotent stem cells inform developmental mechanisms for heart, pancreas, and diaphragm" for consideration by *eLife*. Your article has been reviewed by three peer reviewers, and the evaluation has been overseen by Edward Morrisey as the Senior and Reviewing Editor. The following individual involved in review of your submission has agreed to reveal their identity: Stephen Duncan (Reviewer #2).

The reviewers have discussed the reviews with one another and the Reviewing Editor has drafted this decision to help you prepare a revised submission.

As you will see below, all of the reviewers found your study to have significant merit. In your revisions, please pay particular attention to the request for additional data regarding the role of RA signaling and a needed focus on how *GATA6* regulates cardiac differentiation.

Reviewer #1:

In this manuscript, Sharma et al. performed extensive RNAseq, scRNAseq and ATACseq on early iPS-derived cardiomyocytes with mutation in exon 4 of *GATA6* or deletion of *GATA6*. In addition, they provide an in depth analysis of *GATA6* mutations and their clinical correlates in published data and in the Pediatric Cardiac Genomics consortium. This resulted in an impressive set of data that will be of major use to field and will help in understanding how *GATA6* mutations cause congenital heart disease and pancreas agenesis, among others. This is a major strength of the manuscript. Perhaps the most interesting and unexpected outcome of this work are the data on the *GATA6^R456G/R456G^*mutant, which was generated based on clinical data and in silico modeling indicating an essential role for this amino acid in *GATA6* transcriptional regulation and its role in heart and pancreas development. These data convincingly show enhanced retinoic acid signaling, which is very interesting. This is also the only weakness of the manuscript. With these exciting data in hand, it would perhaps not be too difficult and translationally highly relevant to examine to what extent inhibiting retinoic acid signaling might correct or at least improve the strong in vitro phenotypes observed. No matter the outcome of such experiments, this would be important information for the field.

Reviewer #2:

Review of: *GATA6* Mutations in Induced Pluripotent Stem Cells Inform Developmental Mechanisms for Heart, Pancreas and Diaphragm Malformations – Sharma et al

The authors present an investigation into the perturbed pathways that manifest in some of the most common pathophysiologies associated with *GATA6* haploinsufficiency. The study uses iPSCs to introduce clinically relevant or loss of function mutations into the *GATA6* gene and study the developmental consequences of these using a cardiac differentiation protocol. This study is well-written and offers an interesting and novel insight into the role *GATA6* plays during cardiac development. There is considerable enthusiasm for the significance of the study, but there is concern that the authors do not provide enough evidence to irrefutably justify a subset of the conclusions that they have drawn. This is particularly true of the findings regarding development of pancreas and diaphragm. It would seem that the manuscript would benefit from a greater focus on the cardiac impact of *GATA6* mutations if supported by additional mechanistic studies.

1) *GATA6* ChIP-seq analysis is used from previously published datasets from various endoderm populations and cell lines. However, throughout the manuscript the authors refer to GATA binding motifs, common sites which can be bound to varying degrees by GATA1-6. Overlaying the *GATA6* ChIP-seq with the ATAC-seq data will give greater confidence that *GATA6* does indeed bind to the identified GATA motif. A list is said to be provided in the supplementary data; however, it did not appear to be present in the review package. The binding sites should also be identified in the figures showing individual examples of ATAC-seq genome alignment. However, using ChIP-seq data from a different cell type presents its own concerns. Transcription factors, including GATA4, have been shown to have significantly differential binding patterns between different cell types and also have transient binding sites (PMID: 29358654, PMID: 31350899). Therefore drawing conclusions of direct *GATA6* function using the presence of binding motifs or even ChIP-seq derived from different developmental lineages is problematic. Therefore, ChIP-seq data from the cell type/stage of interest, derived previously, or by the authors themselves, would alleviate this concern.

2) Pancreatic observations are obtained from a cardiac differentiation protocol, using single-cell RNA-seq to identify an endoderm subpopulation. While interesting, these are cells that are being subjected strong exogenous differentiation cues, in a protocol not designed to induce endoderm lineage commitments. Therefore, endodermal differentiation cues are likely being derived via paracrine signaling from mesodermal cells – an undefined variable that may vary between the different *GATA6* genotypes as they follow the developmental trajectories identified. Importantly, *GATA6* haploinsufficiency during pancreatic development using iPSCs has been well-published in the last few years, limiting the novelty of this part of the study (Shi et al., 2017, PMID: 30629940, PMID: 28196690). Indeed, in all three of these studies, FOXA2 has been shown to be down-regulated in *GATA6*^+/-^ and ^-/-^ cells compared to ^+/+^, whereas it is increased in all but the R456G mutant lines in this study. Given the caveats of this differentiation model, and the conserved findings of the published investigations, it would suggest that the model is not optimized for studying endodermal lineages. Consequently, the observations require follow-up using a specific pancreatic differentiation protocol that more tightly controls the factors required for pancreatic development.

3) The following points all relate to the major perceived weakness of this study – several points of interest are identified, but no further investigation of them is attempted. This leaves the paper unfocussed. It would seem that this concern could be addressed by focusing on the cardiac elements of the study if the work were supplemented with mechanistic analyses, rather than trying to generate and partially address such a diversity of hypotheses focusing on other lineages. iPSCs represent a fantastic tool for mechanistic studies, so it is disappointing that the model has not been utilized to its full potential.

4) The authors propose that the *GATA6^R456G/R456G^* line has differential transcriptional activity due to altered capacity for DNA binding. The changes in transcriptional and accessibility profile support this hypothesis, but do not confirm it. *GATA6* ChIP-seq should be performed to elucidate the differential *GATA6* binding that occurs in *GATA6*^+/+^ and *GATA6^R456G^* cells. If ChIP-seq is not possible, CHIP-PCR at a number of sites with differential accessibility highlighted throughout the manuscript is required.

5) Previously derived Hi-C analysis is used to establish potential interactions between distal lncRNA and the HAND2 gene, with GATA-binding motifs found at sites of interaction, and reduced chromatin accessibility observed in the *GATA6* mutant cells. Localized 3-C or equivalent analysis in the *GATA6*^+/+^ and mutant cells should be used to confirm whether the interactions are perturbed, as hypothesized.

6) Reduced chromatin accessibility around PLUT1 is identified as a potential cause of the reduced PDX1 expression seen in the endodermal sub-population. The expression of PLUT1 was undetectable using the RNA-seq datasets. Therefore, RT-qPCR should be used to confirm PLUT1 expression is altered in *GATA6* mutant cells. This investigation would preferably be completed in a specific pancreatic differentiation model, for the reasons outlined above.

7) Increased retinoic acid (RA) signaling is a proposed mechanism by which *GATA6*^R456G^ mutant cells induce aberrant changes to accessibility changes to induce malformation of the diaphragm. Attempts to rescue this phenotype should be made using RA inhibitors, or phenocopy the observed changes in *GATA6* (^+/+^) cells by supplementing with RA.

Reviewer #3:

Overall this is an excellent set of studies from labs that have an established expertise in the definition of the mechanisms that contribute to CHD. These studies clearly link the clinical genotypic and phenotypic data with mechanisms that are further explored using hiPSC-derived CMs. The manuscript is well written with excellent figures and controls (comparing *Gata6* WT, Het, and Variants). The following major issues are intended to help make an already robust study better.

1) Figure 4: The scRNA-esq analysis focused on a few (limited selected) markers. The authors need to use unbiased pathway analysis to characterize the molecular programs of markers specifically expressed in four clusters identified in D4 and five clusters from D8. A comparison with published scRNA-seq of in vivo heart development (e.g. de Soysa et al., 2019) would be a better way to comprehensively characterize the impact of *Gata6* mutations on the developmental program of second heart field.

2) Figure 5: The ATAC-seq analysis was also limited on *HAND2* and *SMYD1*. It would be interesting to see the global changes of chromatin accessibility landscape among the four *GATA6* conditions, by using tools such chromVAR.

As the reviewers were unable to identify CMs in the absence of *Gata6*, is it possible that in this context *Gata6* is functioning as a pioneer factor? Based on the ATAC-seq dataset and the ChIP-seq dataset does *Gata6* bind nucleosomal DNA?

---

## [Author Response]

Reviewer #1:In this manuscript, Sharma et al. performed extensive RNAseq, scRNAseq and ATACseq on early iPS-derived cardiomyocytes with mutation in exon 4 of GATA6 or deletion of GATA6. In addition, they provide an in depth analysis of GATA6 mutations and their clinical correlates in published data and in the Pediatric Cardiac Genomics consortium. This resulted in an impressive set of data that will be of major use to field and will help in understanding how GATA6 mutations cause congenital heart disease and pancreas agenesis, among others. This is a major strength of the manuscript. Perhaps the most interesting and unexpected outcome of this work are the data on the GATA6^R456G/R456G^ mutant, which was generated based on clinical data and in silico modeling indicating an essential role for this amino acid in GATA6 transcriptional regulation and its role in heart and pancreas development. These data convincingly show enhanced retinoic acid signaling, which is very interesting. This is also the only weakness of the manuscript. With these exciting data in hand, it would perhaps not be too difficult and translationally highly relevant to examine to what extent inhibiting retinoic acid signaling might correct or at least improve the strong in vitro phenotypes observed. No matter the outcome of such experiments, this would be important information for the field.

Thank you for this positive impression of our manuscript. We were delayed due to the COVID sequestration but have now performed retinoic acid inhibition experiments in wild type and *GATA6^R456G/R456G^* cells at day 4 of differentiation. We treated cells for 24 hours with DMSO or WIN 18446 (Tocris), an inhibitor of ALDH1A2 that catalyzes the synthesis of retinoic acid from retinaldehyde. From RNA-seq analyses of treated WT hiPSCs we observed downregulation of transcripts associated with heart and muscle cell differentiation and increased expression of transcripts associated with endodermal fate specification. By contrast, treated *GATA6^R456G/R456G^*hiPSCs had increased expression of cardiovascular genes, including *HAND2*. Overall, 25% of transcripts in treated *GATA6^R456G/R456G^*hiPSCs were normal or normalized toward levels found in untreated WT cells, including retinoic acid signaling genes (*HOXA1, HOXB1*). These data are included in revised Figure 4—figure supplement 1.

Reviewer #2:Review of: GATA6 Mutations in Induced Pluripotent Stem Cells Inform Developmental Mechanisms for Heart, Pancreas and Diaphragm Malformations – Sharma et alThe authors present an investigation into the perturbed pathways that manifest in some of the most common pathophysiologies associated with GATA6 haploinsufficiency. The study uses iPSCs to introduce clinically relevant or loss of function mutations into the GATA6 gene and study the developmental consequences of these using a cardiac differentiation protocol. This study is well-written and offers an interesting and novel insight into the role GATA6 plays during cardiac development. There is considerable enthusiasm for the significance of the study, but there is concern that the authors do not provide enough evidence to irrefutably justify a subset of the conclusions that they have drawn. This is particularly true of the findings regarding development of pancreas and diaphragm. It would seem that the manuscript would benefit from a greater focus on the cardiac impact of GATA6 mutations if supported by additional mechanistic studies.

We appreciate the reviewer’s positive impression of our manuscript. We apologize for the delayed response which reflects both COVID sequestration and requests for many additional experiments that we have performed and incorporated into the revised manuscript. In addition we have substantially reorganized and edited the text to focus on mechanisms for cardiac and extra-cardiac phenotypes.

1) GATA6 ChIP-seq analysis is used from previously published datasets from various endoderm populations and cell lines. However, throughout the manuscript the authors refer to GATA binding motifs, common sites which can be bound to varying degrees by GATA1-6. Overlaying the GATA6 ChIP-seq with the ATAC-seq data will give greater confidence that GATA6 does indeed bind to the identified GATA motif. A list is said to be provided in the supplementary data; however, it did not appear to be present in the review package. The binding sites should also be identified in the figures showing individual examples of ATAC-seq genome alignment. However, using ChIP-seq data from a different cell type presents its own concerns. Transcription factors, including GATA4, have been shown to have significantly differential binding patterns between different cell types and also have transient binding sites (PMID: 29358654, PMID: 31350899). Therefore drawing conclusions of direct GATA6 function using the presence of binding motifs or even ChIP-seq derived from different developmental lineages is problematic. Therefore, ChIP-seq data from the cell type/stage of interest, derived previously, or by the authors themselves, would alleviate this concern.

We agree that ATAC-seq can only provide provisional information regarding transcription factors that regulate gene expression. To more directly define the binding of *GATA6* at GATA motifs we performed *GATA6* ChIP-seq on WT and mutant hiPSCs and re-interpreted ATAC-seq data in the context of these ChIP-seq data. Our new findings are incorporated into the main Figures 5-7 and in supplements associated with each of these figures.

The integration of ChIP-seq and ATAC-seq data led to several new conclusions:

We now show that *GATA6* binds GATA motifs at regions of open chromatin (defined by an ATAC-seq peak) as highlighted in the initial manuscript, and that mutant lines have differential expression of the genes. In addition, we demonstrate that *GATA6* also binds closed chromatin that lacks a GATA binding motif. Further, we describe temporal activation of expression for 36% of genes associated with *GATA6* bound to closed chromatin. Notably, many of these genes encode cardiac transcription factors. With these data we deduced that *GATA6* is as a pioneer factor in cardiac development (Figure 5-6).

Additionally, analyses of *GATA6* ChIP-seq in *GATA6^R456G/R456G^* hiPSCs demonstrate reduced binding to GATA motifs in open chromatin as well as over 3-fold increased binding to closed chromatin (without a GATA motif). Based on these data, we conclude that the missense allele has both loss of function (failed binding of the GATA motif) and gain of function (increased binding of closed chromatin, and aberrant gene expression) activities. Together these data help to explain the profound shift in gene expression observed in missense cells.

2) Pancreatic observations are obtained from a cardiac differentiation protocol, using single cell RNA-seq to identify an endoderm subpopulation. While interesting, these are cells that are being subjected strong exogenous differentiation cues, in a protocol not designed to induce endoderm lineage commitments. Therefore, endodermal differentiation cues are likely being derived via paracrine signaling from mesodermal cells – an undefined variable that may vary between the different GATA6 genotypes as they follow the developmental trajectories identified. Importantly, GATA6 haploinsufficiency during pancreatic development using iPSCs has been well-published in the last few years, limiting the novelty of this part of the study (Shi et al., 2017, PMID: 30629940, PMID: 28196690). Indeed, in all three of these studies, FOXA2 has been shown to be down-regulated in GATA6 ^+/-^ and ^-/-^ cells compared to ^+/+^, whereas it is increased in all but the R456G mutant lines in this study. Given the caveats of this differentiation model, and the conserved findings of the published investigations, it would suggest that the model is not optimized for studying endodermal lineages. Consequently, the observations require follow-up using a specific pancreatic differentiation protocol that more tightly controls the factors required for pancreatic development.

Thank you for these thoughtful comments. We fully agree that considerable information has emerged about the how haploinsufficiency and ablation of *GATA6* perturbs endoderm formation. We also agree with the reviewer that the cardiomyocyte differentiation protocol employed here might result in findings than would not occur with endodermal differentiation in vivo. However, we respectfully disagree that the important and novelty of our data is limited due to these considerations. First, to the best of our knowledge, this is the first demonstration of the profound association between *GATA6* exon 4/5 missense alleles and maldevelopment of the pancreas and diaphragm (Figure 1). Missense mutations in other exons do not cause these phenotypes. Second, our data shows that these missense alleles profoundly activate retinoic acid (RA) signaling, and RA signaling is well recognized (see references in the text) to participate in endoderm formation (e.g., RA activation of *HOXA1/HOXB1*) and diaphragm development (RA activation of *STRA6* and *PAX3*). Together these data provide a molecular explanation for the striking occurrence of these malformations with *GATA6* missense mutations in exon 4/5 – erroneous activation of RA signaling. The inclusion of RA inhibition studies confirms these data. The addition of ChIP-seq data provide further support for these conclusions and provide evidence for widespread epigenetic changes when a mutation both depletes *GATA6* and activates RA signaling, as occurs in the missense lines. Together our data demonstrate the critical importance of the zinc finger-2 binding domain in *GATA6*. Interactions by this domain are essential for cardiac, pancreas and diaphragm development. Finally, understanding why some but not all missense alleles cause these extra-cardiac malformations is clinically important for appropriate recognition of the potential developmental outcomes associated with *GATA6* mutations.

3) The following points all relate to the major perceived weakness of this study – several points of interest are identified, but no further investigation of them is attempted. This leaves the paper unfocussed. It would seem that this concern could be addressed by focusing on the cardiac elements of the study if the work were supplemented with mechanistic analyses, rather than trying to generate and partially address such a diversity of hypotheses focusing on other lineages. iPSCs represent a fantastic tool for mechanistic studies, so it is disappointing that the model has not been utilized to its full potential.4) The authors propose that the GATA6 ^R456G/R456G^ line has differential transcriptional activity due to altered capacity for DNA binding. The changes in transcriptional and accessibility profile support this hypothesis, but do not confirm it. GATA6 ChIP-seq should be performed to elucidate the differential GATA6 binding that occurs in GATA6 ^+/+^ and GATA6 ^R456G^ cells. If ChIP-seq is not possible, CHIP-PCR at a number of sites with differential accessibility highlighted throughout the manuscript is required.

Thank you for this important point. In addition to adding new experimental data and interpretation, we substantially revised the manuscript to focus on the epigenetic and transcriptional responses to *GATA6* mutations that result in maldevelopment of the heart, pancreas and diaphragm. As detailed above we now show that GATA6 functions as a cardiac pioneer factor that primes chromatin for subsequent transcriptional activation as well as a traditional transcription factor that regulates gene expression. Heterozygous mutations predominantly perturb its function as a pioneer factor, while homozygous deletion also abrogates functions related to transcriptional activation.

The comparative GATA6 ChIP-seq data from WT and mutant cells are illustrative of how molecular changes elicit clinical phenotypes. We observe ~17,000 ChIP-seq peaks in *GATA6^+/-^* cells, with ~9,000 downregulated and ~6,000 upregulated when compared to WT. The vast majority of associated transcriptional changes indicate decreased gene expression. By contrast there are 21,000 GATA6 ChIP-seq peaks in WT but over 65,000 ChIP-seq peaks in *GATA6^R456G/R456G^* cells, with most peaks residing in areas of closed chromatin. These are associated with widespread abnormal transcriptional profiles (increased and decreased in comparison to WT) in *GATA6^R456G/R456G^* cells that are not found in *GATA6^+/-^* cells. These data has been incorporated into a new set of figures (e.g. Figures 5-6 with associated supplementary figures and files).

5) Previously derived Hi-C analysis is used to establish potential interactions between distal lncRNA and the HAND2 gene, with GATA binding motifs found at sites of interaction, and reduced chromatin accessibility observed in the GATA6 mutant cells. Localized 3-C or equivalent analysis in the GATA6 ^+/+^ and mutant cells should be used to confirm whether the interactions are perturbed, as hypothesized.

Our GATA6 ChIP-seq results identified some GATA6 binding to regions discovered by Hi-C. However, these reads are not statistically enriched over input, implying indicate that the Hi-C interactions are unlikely to have major regulatory effects on *HAND2* expression. We have therefore removed this data from the revised manuscript. However, the GATA6 ChIP-seq does identify direct GATA6 binding near the *HAND2* locus. As these closed chromatin binding sites are altered in GATA6 mutants, we are still confident that GATA6 participates in regulating *HAND2* gene, through its functions as a pioneer factor (see Figure 7A).

6) Reduced chromatin accessibility around PLUT1 is identified as a potential cause of the reduced PDX1 expression seen in the endodermal sub-population. The expression of PLUT1 was undetectable using the RNA-seq datasets. Therefore, RT-qPCR should be used to confirm PLUT1 expression is altered in GATA6 mutant cells. This investigation would preferably be completed in a specific pancreatic differentiation model, for the reasons outlined above.

While our original hypothesis suggested that PLUT1 was a potential cause of reduced *PDX1* expression at day 4, our GATA6 ChIP-seq analysis found that *GATA6^R456G/R456G^* was aberrantly bound to the *PDX1* locus. WT cells do not show GATA6 bound to this locus. At Day 8, GATA6 is bound at the *PDX1* locus, and the ChIP-seq peak is not present in *GATA6^+/-^* or *GATA6^-/-^* cells, providing a potential PLUT1-independent mechanism of *PDX1* regulation. This data is shown in Figure 7—figure supplement 1.

7) Increased retinoic acid (RA) signaling is a proposed mechanism by which GATA6 ^R456G^ mutant cells induce aberrant changes to accessibility changes to induce malformation of the diaphragm. Attempts to rescue this phenotype should be made using RA inhibitors, or phenocopy the observed changes in GATA6 (^+/+^) cells by supplementing with RA.

Reviewer 1 also proposed this idea. To address these comments we treated WT and missense cells for 24 hours with DMSO or WIN 18446 (Tocris), an inhibitor of ALDH1A2, which catalyzes the synthesis of retinoic acid from retinaldehyde. From RNA-seq analyses of WT hiPSCs we observed downregulation of transcripts associated with heart and muscle cell differentiation and increase expression of transcripts associated with endodermal fate specification. By contrast, treated *GATA6^R456G/R456G^*hiPSCs had increased expression of cardiovascular genes, including *HAND2*. Overall, 25% of transcripts in treated *GATA6^R456G/R456G^*hiPSCs were normal or normalized toward levels found in untreated WT cells, including retinoic acid signaling genes (*HOXA1, HOXB1*). These data are included in revised Figure 3—figure supplement 2. Please also see Figure 4—figure supplement 1.

Reviewer #3:Overall this is an excellent set of studies from labs that have an established expertise in the definition of the mechanisms that contribute to CHD. These studies clearly link the clinical genotypic and phenotypic data with mechanisms that are further explored using hiPSC-derived CMs. The manuscript is well written with excellent figures and controls (comparing Gata6 WT, Het, and Variants). The following major issues are intended to help make an already robust study better.

We appreciate the reviewer’s positive impression of our manuscript. We apologize for the delayed response which reflects both COVID sequestration and requests for many additional experiments that we have performed and incorporated into the revised manuscript.

1) Figure 4: The scRNA-esq analysis focused on a few (limited selected) markers. The authors need to use unbiased pathway analysis to characterize the molecular programs of markers specifically expressed in four clusters identified in D4 and five clusters from D8. A comparison with published scRNA-seq of in vivo heart development (e.g. de Soysa et al., 2019) would be a better way to comprehensively characterize the impact of Gata6 mutations on the developmental program of second heart field.

We appreciate this suggestion. We were very aware of the elegant study by de Soysa and colleagues, although we concluded that these mouse data (starting as E7.75) may not be directly relevant to our study that is focused on human development. We also note that there is very limited single cell RNAseq data derived from few captured human cells at gestational age of 7 weeks, which corresponds to E10.5 of mouse hearts (Cui et al., Cell Reports 2019). Our analyses of iPSCs as they begin lineage commitment, identified transcriptional changes by day 4, a time that likely precedes developmental stages characterized in the published single cell mouse and human studies. Additionally as pointed out in the Introduction, GATA6 mutations engineered into mice do not recapitulate phenotypes found in humans. We also hope that the reader recognizes that we did perform unbiased pathway analyses, which led to the Figure 3 in the manuscript. We provide these for the reviewer’s interest in Author response image 1. They are not incorporated into the text due to the considerable information already provided and because of concern by other reviewers that study was insufficiently focused.

**Author response image 1. sa2fig1:** Gene ontology analysis of altered gene networks in day 8 GATA6 LoF and R456G missense cells. Gene ontology analysis comparing wild type to GATA6 LoF or R456G cells. Downregulated cardiac gene networks of interest highlighted in red. Upregulated neurodevelopmental gene networks of interest highlighted in blue. Upregulated EMT gene networks of interest highlighted in green. Analysis conducted using gProfileR package in R and uploaded to REVIGO online software (http://revigo.irb.hr/) to generate plots of top 20 GO pathway terms by p value.

2) Figure 5: The ATAC-seq analysis was also limited on HAND2 and SMYD1. It would be interesting to see the global changes of chromatin accessibility landscape among the four GATA6 conditions, by using tools such chromVAR.

A broader analysis of the ATAC-seq data is provided in Figure 6—figure supplements 1 and 2. Analysis was performed using HOMER (peak calling and motif analysis) and ChIP-Seeker (genomic localization and annotation). In our opinion, chromVar is a tool that is best suited for single cell ATAC-seq analysis or sparse accessibility data, which does not apply to our dataset.

As the reviewers were unable to identify CMs in the absence of Gata6, is it possible that in this context Gata6 is functioning as a pioneer factor? Based on the ATAC-seq dataset and the ChIP-seq dataset does Gata6 bind nucleosomal DNA?

Thank you for this very important comment. As the reviewer likely recognizes, other GATA proteins have been demonstrated to be pioneer factors. We have now performed extensive analyses of GATA6 ChIP-seq in WT and mutant lines and analyzed these data in the context of ATAC-seq data and temporal expression of genes. We find that in WT cells GATA6 primarily binds closed chromatin (nucleosomal DNA), as only ~10% of the ChIP-seq peaks overlap with ATAC-seq data. We identified temporal increases in gene expression among 36.4% of genes associated with GATA6-bound closed chromatin. Among these differentially expressed genes, 56% encode key cardiac developmental transcription factors, *GATA4, SMYD1, KDR,* and *TBX5* (See Figure 5—figure supplement 1 and Supplementary file 5). Based on these data, we propose that GATA6 is a cardiac development pioneer factor. Moreover, we find that the pioneer factor functions of GATA6 are dysregulated in *GATA6^-/+^* cells and even more so in *GATA6^R456G/R456G^* cells These new data are incorporated into Figures 5-7.